# Tuning magnetoresistance in molybdenum disulphide and graphene using a molecular spin transition

Subhadeep Datta[1,2], Yongqing Cai[3], Indra Yudhistira[4], Zebing Zeng[2], Yong-Wei Zhang[3], Han Zhang [1], Shaffique Adam [4,5], Jishan Wu [2] & Kian Ping Loh [1,2]

Coupling spins of molecular magnets to two-dimensional (2D) materials provides a framework to manipulate the magneto-conductance of 2D materials. However, with most molecules, the spin coupling is usually weak and devices fabricated from these require operation at low temperatures, which prevents practical applications. Here, we demonstrate field-effect transistors based on the coupling of a magnetic molecule quinoidal dithienyl perylenequinodimethane (QDTP) to 2D materials. Uniquely, QDTP switches from a spin-singlet state at low temperature to a spin-triplet state above 370 K, and the spin transition can be electrically transduced by both graphene and molybdenum disulphide. Graphene-QDTP shows hole-doping and a large positive magnetoresistance (~50%), while molybdenum disulphide-QDTP demonstrates electron-doping and a switch to large negative magnetoresistance (~100%) above the magnetic transition. Our work shows the promise of spin detection at high temperature by coupling 2D materials and molecular magnets.

[1] SZU-NUS Collaborative Innovation Centre for Optoelectronic Science & Technology, and Key Laboratory of Optoelectronic Devices and Systems of Ministry of Education and Guangdong Province, College of Optoelectronic Engineering, Shenzhen University, Shenzhen 518060, China. [2] Department of Chemistry and Centre for Advanced 2D Materials (CA2DM), National University of Singapore, 3 Science Drive, Singapore 117543, Singapore. [3] Institute of High Performance Computing, A*STAR, Singapore 138632, Singapore. [4] Department of Physics, Centre for Advanced 2D Materials (CA2DM), National University of Singapore, Singapore 117551, Singapore. [5] Yale-NUS College, 16 College Avenue West, Singapore 138527, Singapore. Subhadeep Datta and Yongqing Cai contributed equally to this work. Correspondence and requests for materials should be addressed to J.W. (email: chmwuj@nus.edu.sg) or to K.P.L. (email: chmlohkp@nus.edu.sg)

Hybrid electronic devices containing magnetic molecules are attractive, as data storage units in quantum information processing, as it enables electric-field controlled spin manipulation[1, 2]. However, for an efficient read-out of the spin state of the magnetic molecule, it is important to optimize the magnetoconductance (MC), which is the change in electrical conductance under an external magnetic field, since this parameter characterizes the coupling of the spin state to the local electronic environment[3]. The electronic coupling between a non-magnetic conductor and the magnetic molecule is used to electrically detect the spin polarization: ideally, this coupling can give rise to a giant magnetoconductance effect that can be electrically transduced. For example, quantum dots directly connected to two metallic or superconducting contacts (source-drain electrode) and capacitively coupled to a third gate electrode, can be used as spin valves; the electrical currents are influenced by the high spin state of the weakly coupled molecular magnet (SMM) crystals physically coupled to the quantum dots[1, 4, 5]. However, such a device performs only at extremely low temperatures (below 4 K) and hence, these materials have not been applied so far in practical devices[4–8]. It is important therefore, to develop magnetic nanomaterials with a room temperature spin state that can be coupled with a suitable electrical conductor via electrostatic, dipolar or flux coupling to detect the spin polarization.

In this context, polycyclic aromatic hydrocarbons with an open-shell singlet diradical ground state have been investigated for potential applications in organic spintronics[9]. Recently, Zeng et al. reported a molecule quinoidal dithienyl perylenequinodimethane (QDTP) (Fig. 1a), that, while existing as a singlet diradical in the ground state, is in equilibrium with a triplet biradical that is thermally excited around 360 K[10] (for SQUID measurement, Fig. 1b). Thus, QDTP, with a spin transition at room or higher temperature, has a clear advantage over magnetic nanoparticles or SMMs, and therefore, is well adapted for use in spintronic devices. To electrically detect these molecules, it is necessary to introduce a spin detector, for which a two-dimensional (2D) atomic thin layer that has a high sensitivity to spin states arising either from orbital hybridization effects or from charge transfer, is an ideal candidate[10].

Recently, electrical transport experiments conducted at room temperature have identified a number of advantages of using 2D atomic layers, such as long spin diffusion length, large spin signal, and relatively long spin lifetime as compared to metals (Cu, Ag, or Al) or semiconductors (heavily doped Si, GaAs, or Ge)[11]. Moreover, 2D atomic layers have also been exploited as non-magnetic spacers in the conventional giant magnetoresistive or tunneling magnetoresistive multilayer devices and have shown greater potential for enhancing spin polarization as compared with oxide films[12, 13].

Until now, experimental and theoretical investigations of the role of disorder in magnetoconductance have focussed mainly on enhancing the spin-orbit coupling strength of pristine graphene by introducing adatoms, such as heavy metal atoms or atomic hydrogen[14–17]. In contrast, the effective coupling of room-temperature stable molecular spins and graphene's magnetoconductance allows one to study spin state-dependent electron transport, which can form the basis of a high-temperature magnetoelectronic device[18, 19]. On the other hand, $MoS_2$, a semiconducting 2D transition metal dichalcogenide, provides a good platform for studying the role of Coulomb interaction in molecular spin detection. A large electron interaction strength (three orders larger than graphene) due to reduced dielectric screening in semiconducting 2D crystals and the relatively heavy electron band masses of Mo d-orbitals in the presence of high carrier densities indicates that $MoS_2$ should be an ideal system for studying carrier–spin interaction[20].

Here we demonstrate a hybrid device comprising of a layer of QDTP molecules, which are charged coupled to 2D materials such as graphene or $MoS_2$. We have chosen graphene as a channel material due to its semimetallic band structure and an intrinsically low-spin-orbit coupling, which allows the study of long spin coherence. The MC at high temperatures (300–390 K) of these two hybrid systems, QDTP-graphene, and QDTP-$MoS_2$, are systematically investigated. Spin transition from the singlet to the triplet state ( ~ 360 K) enhances MC in the hybrid devices. Different charge transfer behavior (p or n-type, respectively) is observed for graphene and $MoS_2$ due to the difference in how QDTP molecules bind to these substrates. On graphene, we observe the creation of inhomogeneous charge impurities that lead to an increase in resistance with magnetic field, whereas, the exchange interaction between the triplet spin state of the molecule and $MoS_2$ results in a

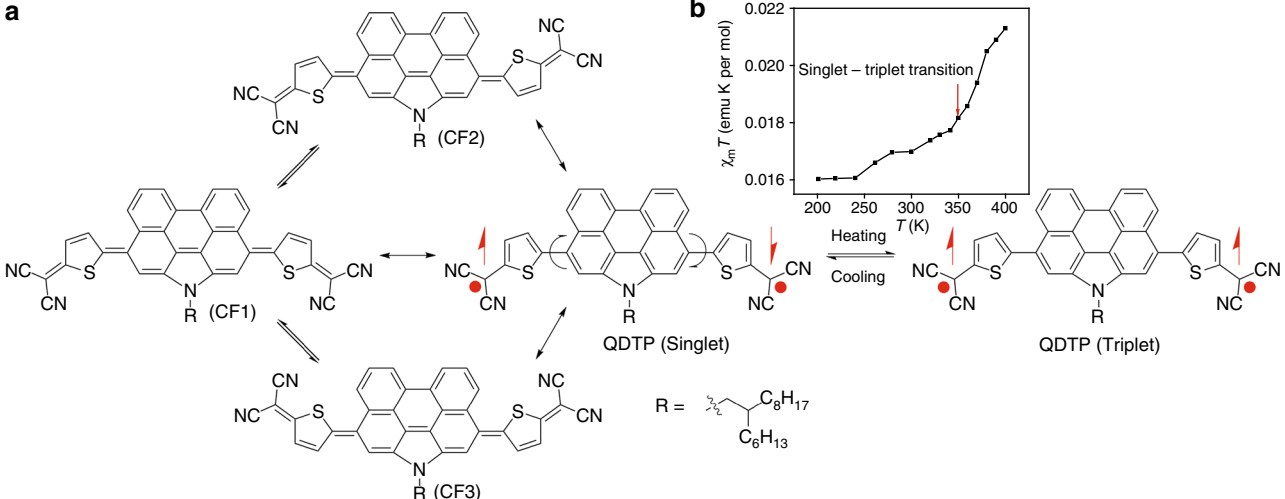

**Fig. 1** The 2D-spin hybrid. **a** Chemical structure of QDTP with schematic representation of singlet to triplet spin transition at 360 K. Quinoidal QDTP exists in three conformers, CF1, CF2, and CF3. The *red dots* are radicals, and the up and down *arrows* indicate spin up and down, respectively **b** SQUID measurements on the QDTP samples presented in the manuscript. The data shows a clear spin transition (increase in spin signal $\chi T$) around 360 K, as indicated by an *arrow*

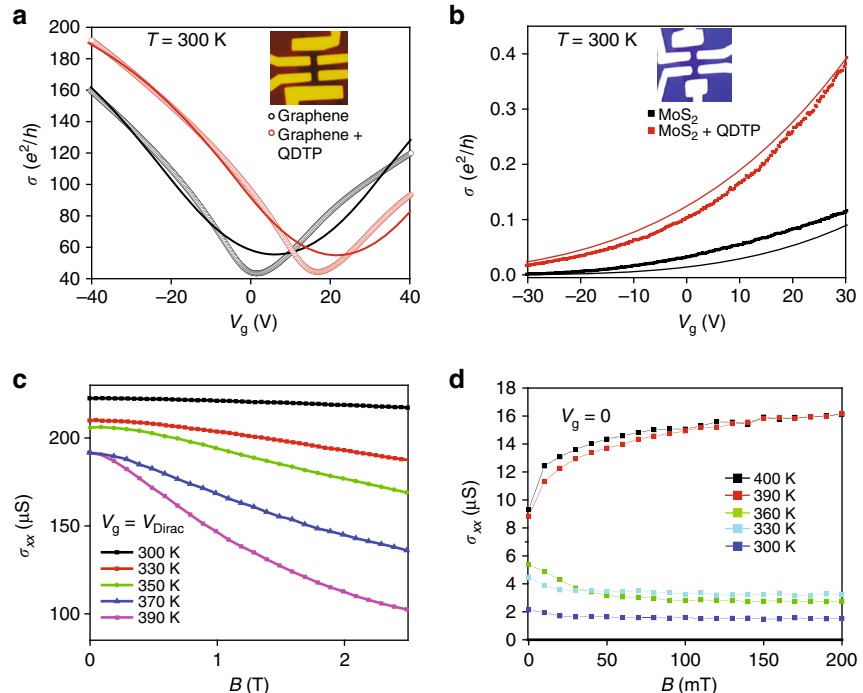

**Fig. 2** Transfer characteristics and longitudinal magnetoconductivity of 2D hybrid devices. **a** Gate dependence of the two terminal conductance at room temperature graphene field-effect transistors (FET) ($\mu \sim 23200$ cm$^2$ V$^{-1}$ s$^{-1}$); pristine graphene (*black*) and after deposition of QDTP molecule (*red*). *Solid lines* represents the theoretical fit of the experimental data using EMT. A positive shift of charge neutrality point $V_D$ ($\Delta V_g \sim 17$ V) confirms the $p$-type doping due to molecule. *Inset*: Typical Hall bar graphene—QDTP hybrid device. **b** Room-temperature transfer characteristic for MoS$_2$ FETs; pristine MoS$_2$ (*black*) and after deposition of QDTP molecule (*red*). *Solid lines* represents the power law fit ($\sigma \sim (V_g - V_0)^\beta$, $\beta$ here is 3.80 and $V_0$ is the threshold voltage of the experimental data. Increment of conductance after the deposition of QDTP molecule confirms the $n$-type doping of MoS$_2$. **c** Longitudinal magneto-conductivity ($\sigma_{xx}$) at charge neutrality point for different temperatures (300–390 K) for graphene-QDTP. Zero-field values of $\sigma_{xx}$ decreases with increasing temperature. The *solid lines* show the fits using equation for $\sigma_{xx}$ ($B$). **d** In the case of MoS$_2$ hybrid, zero-field values of $\sigma_{xx}$ increases with temperature. Positive magnetoconductivity observed above the molecular spin transition temperature (360 K)

sharp transition of magnetoconductance ($\sim 100\%$) at the spin transition temperature.

## Results

**Fabrication**. Monolayers of graphene or MoS$_2$ were micro-mechanically exfoliated from the respective bulk crystals using standard scotch tape method and mounted on degenerately doped silicon substrates with a 300-nm-thick SiO$_2$ overlayer that served as the gate dielectric. The monolayer films were characterized by photoluminescence (PL) (Supplementary Fig. 1), Raman spectroscopy and atomic force microscopy (AFM) (Supplementary Fig. 2). Electrical contacts of Hall-bar-type configuration were fabricated using electron-beam lithography followed by the evaporation of 3 nm of titanium (Ti) and 75 nm of gold (Au). The device was annealed at 420 K to remove resist residues and to reduce contact resistance (see METHODS). The magnetconductance ($\sigma_{xx}$) and Hall conductivity ($\sigma_{xy}$) were measured as a function of temperature ($T$) and magnetic field ($B$) using Physical Property Measurement System from Quantum Design (PPMS). Applying a gate voltage $V_g$, to the Si substrate controlled the charge density in the active channel (graphene or MoS$_2$) device.

QDTP molecules (Fig. 1a) dispersed in dichloromethane (DCM) were drop casted on the devices followed by annealing under vacuum at 400 K to remove the solvent (see METHODS). Atomic force microscopy (AFM) confirmed the successful attachment of the molecule on graphene/MoS$_2$ (Supplementary Fig. 2a, b). In the following two sections, we will discuss the electrical transport at high temperature in graphene and MoS$_2$ based hybrid devices.

**Transfer characteristics**. Figure 2a shows the typical two-probe room temperature (RT) transfer characteristics ($\sigma$–$V_g$ plot) of pristine graphene field effect transistor, which reveals an ambipolar behavior with Dirac point at $V_g = 1.0$ V. A positive shift ($\Delta V_g \sim 17$ V) of the charge neutrality point ($V_D$) after the deposition of QDTP indicates $p$-type doping by the molecule (for detailed statistics see Supplementary Figs. 4a and 5a, and Supplementary Table 1). The experimental data $\sigma(V_g)$ can be fitted well by considering effective medium theory (EMT)[14] and using fitting parameters such as carrier mobility $\mu$, and fluctuation of carrier density $n_{r.m.s.}$, which are used to calculate the Boltzmann–RPA conductivity due to screened Coulomb scattering. This Boltzmann–RPA conductivity $\sigma_B[n, \mu]$ enters into zero magnetic field EMT equation $\int_{-\infty}^{\infty} dn\, P[n, n_0, n_{r.m.s.}] \frac{\sigma_B - \sigma_{EMT}}{\sigma_B + \sigma_{EMT}} = 0$, where $P[n, n_0, n_{r.m.s.}]$ is a Gaussian distribution centered at an average carrier density $n_0$ with an r.m.s. fluctuation given by $n_{r.m.s.}$. For QDTP-graphene hybrid, we have used an additional parameter, i.e., the short-range conductivity $\sigma_S$ for Boltzmann–RPA conductivity, which captures the saturation of conductivity at high carrier density. The carrier mobility $\mu = (23200 \pm 1000)$ cm$^2$ V$^{-1}$ s$^{-1}$ and fluctuation of carrier density $n_{r.m.s.} = (118 \pm 8)10^{10}$ cm$^{-2}$ for pristine graphene and $\mu = (25000 \pm 1000)$ cm$^2$ V$^{-1}$ s$^{-1}$, $n_{r.m.s.} = (109 \pm 8)10^{10}$ cm$^{-2}$ and $\sigma_s = (335 \pm 23)e^2/h$ for graphene-QDTP hybrid can be obtained after careful fitting. Experiments carried out on graphene flake with relatively smaller mobility (2800 cm$^2$ V$^{-1}$ s$^{-1}$) also shows qualitatively similar behavior (mobility of hybrid device $\mu \sim 6400$ cm$^2$ V$^{-1}$ s$^{-1}$) after deposition of QDTP molecules (Supplementary Fig. 5a). Increment of carrier mobility in

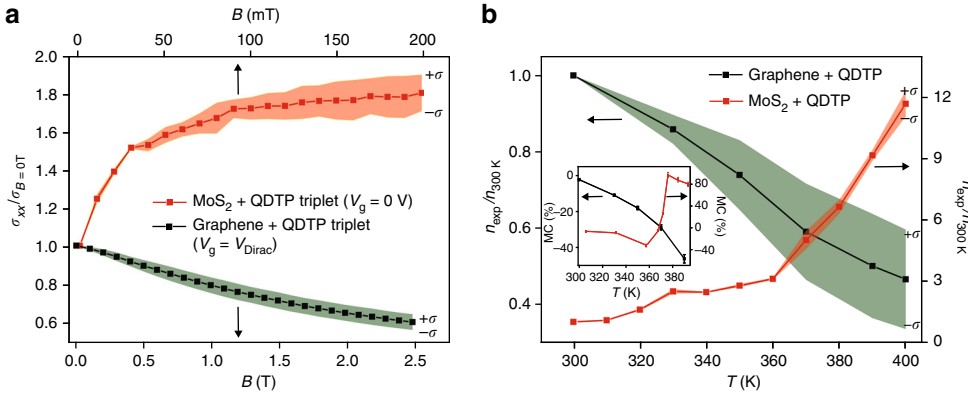

**Fig. 3** Magnetoconductance and carrier density of 2D hybrid devices. **a** The mean of normalized magneto-conductivity with respect to zero-field value ($\sigma_{xx}(B)/\sigma_{xx}(B=0)$) at 390 K (spin triplet state of QDTP) for 10 different samples from each hybrid has been plotted to show the consistency of the result. The colored region (*red* for MoS$_2$ and *green* for graphene) depicts the s.d. ($\pm\sigma$) of the data for each magnetic field. Graphene–QDTP hybrid (*black*) shows large negative differential magneto-conductivity ($\Delta\sigma_{xx}|_{390\,K} = -2.5\,e^2/h$) at 390 K, compared with room temperature. In contrast, MoS$_2$–QDTP hybrid (*red*) shows large positive differential magneto-conductivity ($\Delta\sigma_{xx}|_{390\,K} = +0.23\,e^2/h$) at 390 K, compared with room temperature. **b** Mean of normalized carrier density ($n_{exp}/n_{300K}$) as a function of temperature (300–390 K). S.D. for each data point has been shown in colored filling ($\pm\sigma$). Graphene–QDTP hybrid (*black*) shows a gradual decrease of carrier density in the given temperature window from its room temperature value ($n_{390\,K}/n_{300\,K} \sim 0.5$), calculated from the theoretical fit of raw $\sigma_{xx}(B)$ data, shown in Supplementary Fig. 5, and Supplementary Table 3. In contrast, MoS$_2$–QDTP hybrid (*red*) shows a sharp increase of carrier density at 370 K (spin transition temperature of QDTP), and reaches 10 times larger value at 400 K from its room temperature value. *Inset*: Magnetoconductance (MC) of the two hybrid devices. Graphene—QDTP hybrid (*black*) shows a negative MC of 50%, while MoS$_2$ – QDTP hybrid (*red*) shows large positive MC of 100% with a sharp jump around 370 K

graphene after the deposition of QDTP molecule can be explained by compensation of charged impurities, as has been reported in the previous studies of nanoparticles coated graphene[21, 22]. From the difference of $V_D$, we get the concentration of holes introduced by molecule for the two devices as $\Delta p = 252 \times 10^{10}$ cm$^{-2}$ and $105 \times 10^{10}$ cm$^{-2}$, respectively.

The gate-dependent conductivity $\sigma(V_g)$ of pristine MoS$_2$ device at room temperature (Fig. 2b) are similar to that reported in the literature[23]; initially, $\sigma(V_g)$ rises slowly with increasing back gate voltage, followed by a rapid rise in current as the Fermi level approaches the bottom of the conduction band. For the hybrid device, doping with QDTP leads to an enhancement of $\sigma(V_g)$ by a factor of three from its pristine value ($\sigma^{Doped} \sim 0.10\,e^2/h$ vs. $\sigma^{Pristine} \sim 0.03\,e^2/h$ at $V_g = 0$). Qualitatively similar effect occurs ($\sigma^{Doped} \sim 0.06\,e^2/h$ vs. $\sigma^{Pristine} \sim 0.02\,e^2/h$ at $V_g = 0$) for other device which has lower mobility. Inspite of the fact that the conductivity of both pristine MoS$_2$ and MoS$_2$-QDTP hybrid is less than $\frac{e^2}{h}$, it follows a power-law gate-dependence in the metallic regime: $\sigma = A(V_g - V_0)^\beta$, where $A$ is the global parameter, $\beta$ is the critical exponent, and $V_0$ is local parameter. From the difference of $V_0(V_0|_{MoS_2} = -28.5\,V)$ and $V_0|_{MoS_2+QDTP} = -64.8\,V$, electron doping by QDTP molecule can be extracted as $n = (267\pm1)10^{10}$ cm$^{-2}$ (for detailed statistics see Supplementary Figs. 4b and 5b, and Supplementary Table 2).

**Graphene—QDTP hybrid Hall measurement.** The Hall measurement allows us to study the gate modulation of transverse resistance ($|R_{xy}| = f(V_g)$) at different temperatures ($T$) at a constant perpendicular magnetic field ($B = 2$ T). Increasing the temperature to 390 K, the Dirac point gradually shifts to 28 V, making graphene more p-doped (see Supplementary Fig. 6). The longitudinal conductivity ($\sigma_{xx}$) as a function of magnetic field ($B$) for the temperature window 300–390 K is shown in Fig. 2c. The background magneto-conductance due to the presence of intrinsic defects in pristine graphene is weak ($\Delta\sigma_{xx} \sim -0.1\,\mu$S) and temperature-independent[24] (Supplementary Fig. 7a). Compared with graphene device, the hybrid devices (gate fixed at $V_D$) show two distinct features: First, the zero-field conductivity decreases ($\Delta\sigma_{xx}|_{B=0} = -25\,\mu$S) when the temperature is increased to 390 K,

and second, as a function of the $B$-field, $\sigma_{xx}$ decreases as temperature increases, as shown in Fig. 3c. At 390 K, we recorded a 55% reduction of $\sigma_{xx}$ from its RT value, measured at $B = 2$ T. In addition, Fig. 3a displays the $B$-field dependent longidinal conductivity ratio ($\sigma_{xx}/\sigma_{xx}(B=0)$) of the hybrid device, averaged over 10 samples, measured at the Dirac point at $T = 390$ K ($\sigma_{xx}/\sigma_{xx}(B=0)$ at room temperature in Supplementary Fig. 8a). The observed magneto-conductance, which is consistent in all hybrid devices we have measured, points to the influence of external magnetic moments coupled to the graphene device.

**MoS$_2$–QDTP hybrid Hall measurement.** In the case of MoS$_2$, gate voltage-dependent Hall conductivity ($\sigma_{xy}$) at $B = 2$ T is plotted in Supplementary Fig. 9 in the same temperature window as for graphene. While $\sigma_{xy}$ of pristine MoS$_2$ is temperature independent (Supplementary Fig. 9), the hybrid devices exhibit a steep increase in $\sigma_{xy}$ with temperature; notably, $\sigma_{xy}$ is enhanced by a factor of 7 as the temperature is increased from 300 to 390 K. The similar trend shown in all the measured hybrid devices suggests that first, there is electron doping of MoS$_2$ by the QDTP molecules, and second, the charge transfer from high temperature triplets in the QDTP molecule is greater than that from room temperature singlets. The longitudinal conductivity ($\sigma_{xx}$) of MoS$_2$–QDTP hybrid device as a function of magnetic field ($B$) for the same temperature window as graphene hybrid is shown in Fig. 2d. For pristine MoS$_2$, no change in $\sigma_{xx}(B)$ is observed as the temperature is increased up to 400 K (Supplementary Fig. 7b). However, for the hybrid device at 390 K, $\sigma_{xx}$ increases with applied magnetic field, as shown in Fig. 2d. The differential magneto-conductivity, which is defined by [$\Delta\sigma_{xx} = \sigma_{xx}(B) - \sigma_{xx}(0)$], is slightly negative ($-2\,\mu$S) at 330 K, but increases to 7 $\mu$S at 400 K. The observed increase in zero-field $\sigma_{xx}$ with temperature is consistent with the Hall conductivity ($\sigma_{xy}$) measurement. The magneto-conductivity ($\sigma_{xx}/\sigma_{xx}(B=0)$), averaged over 10 samples, is plotted as a function of $B$-field sweep (0 to 200 mT) at different temperatures from 300 to 400 K in Fig. 3a. Similar to the graphene hybrid, influence of the spin state of magnetic molecule on the electron transport of MoS$_2$ is evident in all the devices,

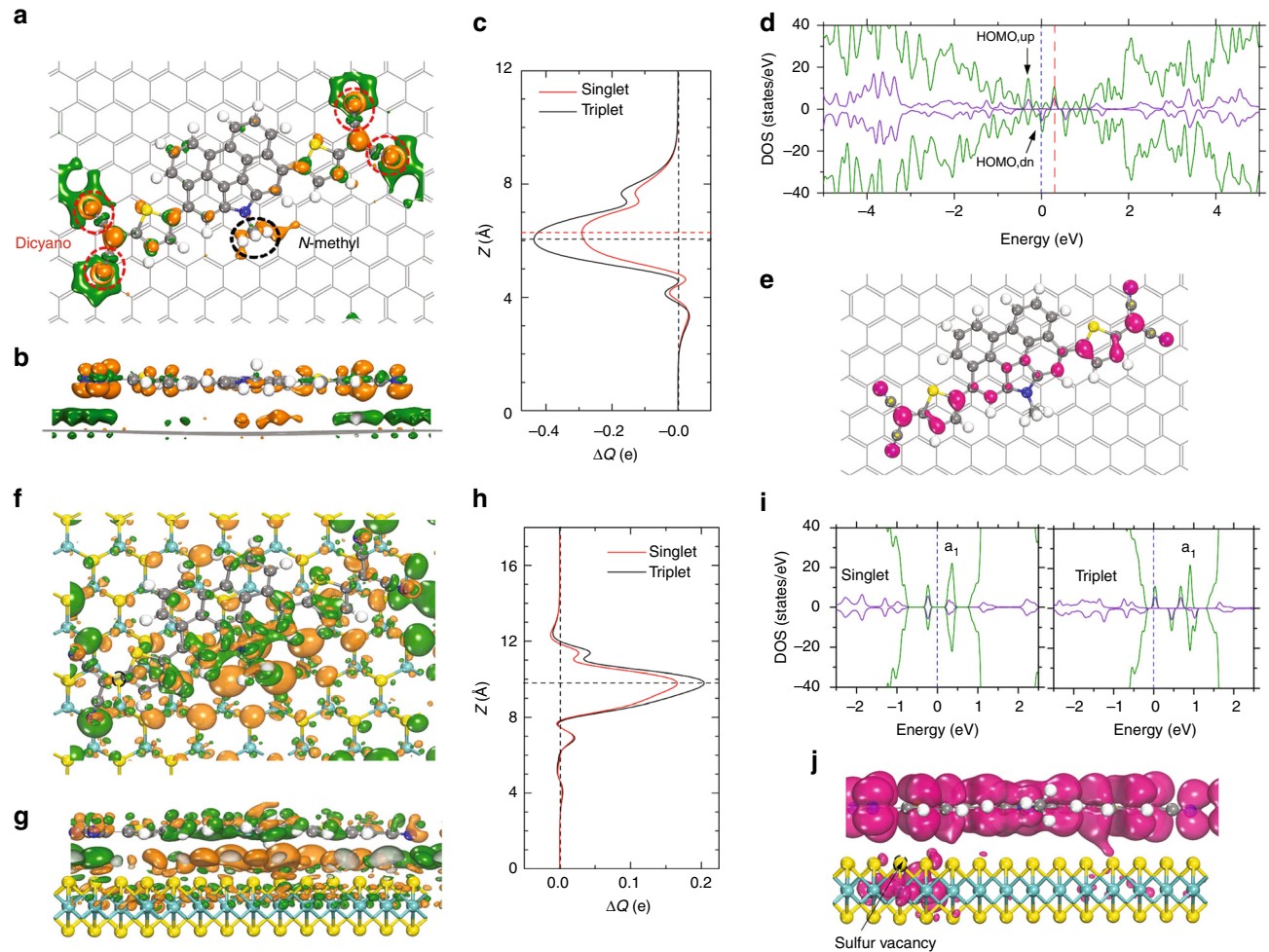

**Fig. 4** Charge transfer between QDTP and graphene/MoS$_2$. Top **a** and side **b** views of the isosurface of differential charge density $\Delta\rho(\mathbf{r})$ at value of the 0.005 Å$^{-3}$ for QDTP (triplet state) adsorbed on graphene. The *green* (*orange*) color denotes loss (accumulation) of electrons. Dicyano groups (*dotted red circle*) of the QDTP molecule act as electron acceptor in the case of graphene–QDTP hybrid. **c** The amount of transferred charge up to the z point $\Delta Q(z)$ for graphene decorated with CF3 QDTP in singlet and triplet states. Partial density of states (PDOS) **d** and spin density **e** for graphene–QDTP hybrid in triplet state. The dashed *blue* (*red*) lines in **d** show the position of Fermi level (Dirac point of graphene). Top **f** and side **g** views of the isosurface of $\Delta\rho(\mathbf{r})$ at value of the 0.005 Å$^{-3}$ for QDTP (triplet state) adsorbed above the sulfur vacancy ($V_S$) site (represented by the *black dashed circle*) of MoS$_2$ with the dicyano group above the $V_S$ center. N–methyl group (indicated in **a** with *dotted black circle*) of QDTP molecule acts as electron donor in the case of MoS$_2$–QDTP hybrid. **h** The amount of transferred charge up to z point $\Delta Q(z)$ for QDTP in singlet and triplet states, adsorbed above a sulfur vacancy ($V_S$) in MoS$_2$. The *horizontal dashed lines* in **c**, **h** define the maximum amount of transferred charge from the molecule to the underlying layers. PDOS **i** for MoS$_2$–QDTP hybrid in singlet and triplet states. **j** Spin density for MoS$_2$–QDTP hybrid in triplet state. The green and purple lines in **d**, **i** correspond to the states from the hybrid system and the QDTP molecule, respectively. The $V_S$ related a1 levels are slightly hybridized with the QDTP unoccupied orbital of the spin down component

except for the fact that in the case of MoS$_2$ hybrid, we observe positive magneto-conductance.

**Temperature dependent carrier density**. QDTP-doped graphene can be considered as an impurity-doped system having an inhomogeneous distribution of electron and holes near the Dirac point[25], the magnetoresistivity (or magneto-conductivity) of such system has been analyzed previously using a semi-empirical equation[25]. Herein, the experimental magneto-conductance at Dirac point for each temperature of our graphene-QDTP hybrid is fitted to a modified semi-empirical equation $\sigma_{xx}(B) = \frac{\sigma_{xx,0}}{\left[1+(\mu B)^2\right]^{\frac{1}{2}}} + \sigma_{xx,1}$, as shown in Fig. 2c. The fitting parameter used are high-field conductivity $\sigma_{xx,0}$, low-field conductivity $\sigma_{xx,1}$, and mobility $\mu$. The extra conductivity term ($\sigma_{xx,1}$) in the equation accounts for the imbalance in the electron and hole concentrations, unequal electron and hole mobilities,

and deviation of sample geometry from the ideal Hall bar[26]. Above the transition temperature (370 K), $\sigma_{xx,0}$ (3.36 $e^2/h$) is much larger than $\sigma_{xx,1}$ (1.53 $e^2/h$) indicating that magneto-conductivity is probably better described by the unusual $[1+(\mu B)^2]^{-1/2}$ dependence at higher temperatures (Supplementary Table 3). The good fitting of the experimental data confirms the model of temperature-dependent macroscopic inhomogeneity. The mean carrier density, given by $n(T) = \frac{\sigma_{xx,1}+\sigma_{xx,0}}{e\mu}$, decreases in the given temperature window, as shown in Fig. 3b (shown as normalized carrier density $n_{exp}/n_{300\,K}$, averaged over 10 samples). The temperature dependence of carrier density due to thermal smearing predicted by theory is very small, and should not play an important role in the observed temperature dependence extracted by aforementioned method. This suggests a non-trivial role played by the molecule in carrier activation.

In the case of MoS$_2$, the normalized carrier density ($n_{exp}/n_{300\,K}$), averaged over 10 samples, given by two channel

model $n(T) = \frac{\sigma_{xx}(B\rightarrow 0)}{e\mu}$, increases by an order from the RT ($n_{400K} = 1.2 \times 10^{10}$ cm$^{-2}$), with a sharp change around 360 K (Fig. 3b). The temperature dependence of $\sigma_{xx}$ shows a similar trend as the carrier density increasing 8 times from its RT value.

**Temperature dependent magnetoconductance.** To investigate the origin of high temperature magnetoconductivity in the case of graphene, the gate–voltage dependence of the magnetoconductance (MC$= \frac{\sigma(H)-\sigma(0)}{\sigma(0)} \times 100\%$) is calculated at different temperatures (inset of Fig. 3b). At RT, the MC values are relatively constant at different gate voltages. However, a negative MC of 45% is observed at 390 K, while fixing the gate voltage at Dirac point ($V_D$). In slightly electron/hole doped regions, the change in MC is much lower even though a similar trend is observed (Supplementary Fig. 10). A highly electron doped region is not expected to show a change in MC with temperature, since the fluctuations in carrier density are negligible far away from the Dirac point when compared with the total number of carriers[25]. The enhancement of negative magnetoconductance with temperature is due to the temperature-dependent charge transfer between the QDTP and graphene; at higher temperatures (beyond 360 K), the density of Coulomb impurities increases due to higher charge transfer with the triplet state of the molecule[27]. The MC measured in this hybrid device is relatively large ($\sim 50\%$) in comparison with other disordered-graphene devices ($\sim 20\%$)[28–30], and graphene based vertical spin valves ($\sim 15\%$)[31, 32].

In contrast to graphene, MoS$_2$-QDTP hybrid device shows a positive MC (inset Fig. 3b), which reaches a maximum positive MC value of 80% at 400 K, this is one order of magnitude higher than its RT value of—9%. These results suggests that the carrier density in MoS$_2$ can be tuned by changing the temperature or by external doping and this tunability is the key to efficiently detect the singlet–triplet spin transition in QDTP molecules at high temperatures.

Our electrical transport measurements reveal a strong change in the MC of the hybrid devices at RT vs. 370 K, which may have its origin in the temperature-dependent charge transfer interactions of the respective 2d materials with QDTP molecules. Interestingly, graphene and MoS$_2$ show contrasting evolution in the MC (different signs), which should originate from the intrinsic electrical characteristics of the two materials.

**First principles calculations.** To understand the origin of the differences in the transport properties of hybrid devices of graphene or MoS$_2$ with QDTP, first-principles calculations were performed to understand the charge transfer behavior in these hybrid systems. QDTP has three conformational isomers or conformers (such as CF1, CF2, and CF3), as shown in Fig. 1a. Figure 4 shows the charge transfer between graphene/MoS$_2$ and the adsorbed QDTP molecule. For graphene, the differential charge density analysis shows that the CF3 QDTP accepts a larger number of electrons in the triplet state (0.44$e$ per molecule) than in the singlet state (0.29$e$ per molecule) (Fig. 4c). A similar behavior is observed for the CF1 conformer, albeit with a slightly larger charge transfer than the CF3 conformer (Supplementary Figs. 11–14). The results obtained from these calculations agree with the experimental results, both on the p-type doping of QDTP on pristine graphene, and the gradual positive shift of the Dirac point with temperature due to the enhanced doping in the triplet state. In the case of the hybrid QDTP-MoS$_2$ system, we attribute the enhancement of the carrier density to the interaction between QDTP molecule and the sulfur vacancies ($V_S$) in MoS$_2$. It is well known that sulphur vacancies are ubiquitous in MoS$_2$ and can reach concentrations as high as $10^{10}$ cm$^{-2}$[33]. These defects

have unpaired spins and thus have a stronger interaction with spin molecules. $V_s$ defects also act as charge traps in MoS$_2$, and can be passivated by adsorbates[33], one signature of this is the enhanced photoluminescence (Supplementary Fig. 1) from MoS$_2$ following the adsorption of QTDP molecules. Indeed, based on our spin polarized DFT calculations, the charge transfer between the QDTP and MoS$_2$ plotted in Fig. 4h shows that the triplet-state (0.21$e$ per molecule) adsorption has a much larger charge transfer than that of the singlet-state (0.16$e$ per molecule) adsorption.

Our theoretical calculations show that the different charge transfer behavior observed between graphene and MoS$_2$ has its origin in the different interactions of these 2D layers with the functional groups in the QDTP. The isosurface plot of the differential charge density reveals that the dicyano groups in QDTP act as electron acceptors with respect to graphene (Fig. 4a, b), whereas, the N-methyl (NCH$_3$) group located in the in the mid-section of the molecule acts as an electron donor when present on the surface of MoS$_2$ (Fig. 4f, g).

The local density of states of the spin triplet state for both hybrid systems are shown in Fig. 4d, i, respectively. A tiny amount of spin-polarization is found in the states of graphene and MoS$_2$ due to a weak van der Waals interaction with QTDP. In Fig. 4e, j, our analysis of the spin density shows that significant amount of the magnetic moment is concentrated at the dicyano groups and at the conjugated backbone. As shown by the plot of spin density in Fig. 4j, there is indeed appreciable weight of spin density in the MoS$_2$ sheet. The enhanced spin density in MoS$_2$ is due to the asymmetric coupling of the unpaired electrons in $V_S$ localized states with that of the unpaired spins of the triplet state molecule. Such spin density and asymmetric coupling are absent in the singlet state molecule (Fig. 4i).

**Magnetic field effect in hybrid devices.** The observation of large MC in graphene-QDTP at 400 K rules out any quantum effects, such as weak (anti)localization[28]. In our work, the temperature dependent resistivity near Dirac point is a signature of charged-disorder-dominated transport. The conductivity of an inhomogeneously doped graphene film near the Dirac point is different from the homogenous case, due to correlations in impurity positions. The net effect of charge doping with QDTP is the creation of macroscopic regions of charge puddles; these may have a charge excess or charge deficit with respect to the average value. In the inhomogeneous regime, the conductivity near the Dirac point is dominated by the charge-disorder[27, 34, 35].

To understand the positive MC in case of the MoS$_2$ hybrid, we can consider the magnetic scattering mechanism as reported in diluted magnetic semiconductors (DMS)[36–38], together with spin-imbalance caused by the initialization process of randomly oriented spins[39, 40]. The exchange coupling between the spin triplet state in QDTP and the 2D channel electrons scales linearly with Wigner Seitz radius $r_s$[41]. However, for 2D materials, the value of $r_s$ depends on both intrinsic factors such as the Fermi velocity, and extrinsic factors like the dielectric screening of the substrate[14]. For the carrier densities relevant to our experiment, we estimate that for graphene, $r_s \sim 1$, while for MoS$_2$, $r_s \sim 1000$. We attribute the negative magnetoresistance (positive MC) of MoS$_2$ beyond the triplet transition to Ruderman–Kittel–Kasuya–Yosida (RKKY) coupling of the spin triplet state of QDTP to electron spin in MoS$_2$. This is analogous to the coupling between Mn spin states and Ga electrons in the dilute magnetic semiconductor (Ga,Mn)As[37, 42]. Accordingly, the exchange interaction ($J$) between the high temperature spin triplet state of the QDTP molecule ($S_i$ at site $i$) and the spin of the carrier in MoS$_2$ ($s$) is given by the Hamiltonian, $H = -J\sum_i S_i \cdot s$. This exchange interaction may give rise to the spin-disorder

scattering resistivity[36–38, 43], which can be reduced by $B$-field induced alignment of spins, and can be expressed in a simple form as[37, 43]: $\frac{\rho(B)}{\rho(B=0)} = 1 - A(T) \times B^2$ (see Supplementary Fig. 15 for fitting details). Supplementary Figs. 15 and 16 show the result of the fit using $A(T)$ as temperature dependent fitting parameter which has the same qualitative behavior as temperature dependent magnetic signal data ($\chi T$) from the SQUID measurement (or square of the magnetic moment) of the QDTP molecule[10] (Fig. 1b). The coupling constant $J$ can be extracted from DFT calculations using the energy difference between the spin triplet and singlet states ($\Delta E \sim 0.36$ eV). The theoretical values of $J$ based on the current structural model is estimated to be around 131 eV Å$^{-2}$, which is close to the value (150 eV Å$^3$) reported in (Ga, Mn)As[37]. We have also estimated $J$ to be 200 eV Å$^{-2}$ from the experiment following Supplementary Eq. 1, as discussed in Supplementary Note 1. The observed magnetoconductance is due to the quadratic dependence of exchange coupling constant ($J$) and the spin imbalance prevalent in the QDTP molecule, which is enhanced at high temperature due to the spin flip of the molecule (Supplementary Fig. 20). The sudden increase in MC of MoS$_2$ at the transition temperature confirms the effect of the spin transition of the molecule on the electrical transport of MoS$_2$. In comparison with the earlier reports on Mn-substituted ZnO ($n$-type 3d DMS)[36] or Gd-doped Eu-chalcogenides[38], the positive MC effect in QDTP-doped MoS$_2$ at high temperatures is much larger ( ~ 100%).

## Discussion

Overall, our results show that magnetic molecules on graphene surface can be treated as charge disorder. The transition from singlet to triplet state at high temperature causes a more resistive magneto-transport, which can be enhanced by applying a magnetic field. In the case of MoS$_2$, spin triplets of the adsorbed molecule at high temperature are coupled to charge carriers in MoS$_2$, and an alignment of the non-interacting individual spins with the magnetic field enhances the electron transport. Since the interaction parameter $r_s$ in graphene is much weaker than in MoS$_2$, the effect of exchange interaction with the QDTP molecule is much larger in MoS$_2$. The different magnetoconductance behavior of graphene and MoS$_2$ at the spin transition point of the molecule, which can be exploited to detect the spin flip of the magnetic molecules clearly highlights the importance of using the appropriate conduction channel for different targeted application: semimetallic graphene with a higher carrier density is better tailored for impurity-dominated transport, whereas semiconducting 2d materials such as MoS$_2$ are ideal for detecting spin dependent phenomena.

Our work also highlights the importance of understanding the charge transfer and magnetic phenomena taking place at these molecule-2D hybrid interfaces. Clearly, QDTP molecule enhances the sensitivity of the MoS$_2$ FET to small magnetic field ( < 20 mT) by two orders of magnitude. This suggests that modifying 2D materials with magnetic molecules can enhance the MC by orders of magnitude. The advantage of QTDP is that the enhancement is operational at elevated temperatures, thus allowing the sensor to be operated at cryogen-free conditions. Finally, our work suggests that biradical molecules with active spin centers may be good candidates for room temperature hybrid spintronic devices with multi-functional properties, enabling the sensing of magnetic, electrical, optical, and chemical stimuli.

## Methods

**Device fabrication.** Graphene or MoS$_2$ flake is prepared by mechanical peeling and transferred to heavily doped silicon with a layer of 300 nm thick SiO$_2$ (gate capacitance $C_i = 11.6$ nF/cm$^2$) on top. Optical microscope-based contrast is used to get initial information about the number of layers. To determine the quality and thickness of graphene/MoS$_2$, Raman spectroscopy measurements were carried out. Electrodes were patterned by e-beam lithography using a bilayer resist combination of PMMA and subsequently Ti/Au (3 nm/75 nm) was deposited by an e-beam evaporator. Standard lift-off procedures using warm acetone were followed after the metal deposition. Typical dimension of the channel is as follows: $L = 3$ µm is the channel length, $W = 1.5$ µm is the channel width. QDTP molecules dispersed in DCM were drop casted on the devices followed by annealing under vacuum at 400 K to remove the solvent.

**Computational details.** First-principles calculations based on spin-polarized density functional theory are performed by using Vienna ab initio simulation package[44]. Becke88 optimization (optB88) is adopted for the description of the van der Waals interaction between the physisorbed QDTP and graphene/MoS$_2$ by using a kinetic energy cutoff of 400 eV. Here we consider a single QDTP molecule adsorbed in supercells of graphene and MoS$_2$ containing total atoms of 217 and 183, respectively, with a vacuum layer with a thickness of 15 Å. The first Brillouin zone is sampled with a $2 \times 3 \times 1$ Monkhorst–Pack grid. All the structures are fully optimized by minimizing the forces on each atom less than 0.005 eV Å$^{-1}$. The interfacial charge transfer is obtained via calculating the differential charge density $\Delta\rho(\mathbf{r})$. The amount of transferred electron up to $z$ point in the normal direction is obtained using $\Delta Q(z) = \int_{-\infty}^{z} \Delta\rho(z') dz'$, where $\Delta\rho(z)$ is the plane-average differential charge density through integrating the $\Delta\rho(\mathbf{r})$ in the x–y plane.

**Data availability.** All relevant data are available from the corresponding authors.

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

## Acknowledgements

K.P.L. thanks National Research Foundation, Singapore for NRF Investigator Award: Graphene oxide a new class of catalytic, ionic and molecular sieving materials, award number: NRF-NRF12015. J.W. acknowledges financial support from the MOE Tier 3 programme (MOE2014-T3-1-004) and MOE Tier 2 grant (MOE2014-T2-1-080). S.A. acknowledges financial support from the Singapore National Research Foundation [NRF-NRFF2012-01], Singapore Ministry of Education and Yale-NUS College [Grant No. R-607-265-013121], and the National University of Singapore Young Investigator Award [R-607-000-094-133]. Discussions with Vitor Pereira and Ding Z.J. are gratefully acknowledged. This project is supported by Shenzhen Peacock Plan (Grant No. KQTD2016053112042971).

## Author contributions

K.P.L. supervised the project. The hybrid device was made by S.D. Synthesis of QDTP molecule is done by Z.Z. and J.W. S.D. carried out the measurements. The data analysis was made by S.D. with the help of K.P.L. The theory for graphene hybrid was made by I.Y., S.D. and S.A. The theory for $MoS_2$ was done by S.D. and S.A. DFT simulation was done by Y.C. and Y.-W.Z. S.D. and K.P.L. wrote the manuscript with inputs from H.Z., S.A. and Y.C. All authors discussed the data.

## Additional information

**Competing interests:** The authors declare no competing financial interests.

