## [Peer Review File · Nature Communications]

Reviewers' comments:

Reviewer #1 (Remarks to the Author):

This manuscript reports magnetoresistance experiments on Graphene and MoS₂ samples functionalised with molecules QDTP, which can switch between a spin-singlet to a spin-triplet state above 370K. Results of abinitio calculations investigating the charge transfer between the molecules and the 2D layer are also shown. This work is potentially interesting since it may lead to devices which magnetoresistance can be switched at temperatures above room temperature in a way which depends on a gate voltage. However for the reasons exposed below I think that this work is too preliminary to be published especially in a large impact journal.

1-Samples

The authors present data on 2 different devices one with graphene and one with MoS₂ both before and after deposition of QDTP molecules very different behaviors are found for these 2 systems for the magnetoresistance after deposition of the molecules which are attributed to different charge transfer mechanisms in these 2 systems. Even if it is suggested in the text that several samples were prepared for both systems data are shown for only a single sample of graphene and MoS₂. Previous work on graphene functionalised samples with porphyrins (see Li et al. Phys. Rev. B 93, 045403 (2016)) show that charge transfer and induced magnetism strongly depend on the initial doping state of the sample.

The very broad Dirac point of the graphene sample investigated by the authors indicates a poor quality sample, with probably many dopants on the sample before deposition of the molecules. It would be very important to present data on several better quality samples.

The same comments are valid for the MoS₂ samples. Moreover, deposition of the QTP molecules does not seem to be well controlled. Did the authors investigate different conditions of deposition of the molecules?

2.Data

In order to appreciate the difference between the temperature dependence of the carrier density and magnetoresistance of coated graphene and MoS₂ samples more points in temperature are needed in Fig.3 and of course different samples. Magnetoresistance data is only shown for one direction of the magnetic field. It would have been useful to have also negative fields since an important question is the possible existence of magnetic hysteresis loops in this system.

The authors have to be accurate in their definitions of longitudinal and Hall conductivity. They are both called ρ_{xx} (line 101). Moreover the curve shown in S2 is very strange since it is very unlikely that the Hall resistance goes through a minimum at the Dirac point. It should instead change sign !. "Transverse transfer curve" in the title of the figure caption does not mean anything.

3.Interpretation of the magnetoresistance data

The authors suggest that the difference between the magnetoresistance of coated graphene and MoS₂ samples is due to larger RKKY interactions between the molecules in the triplet state on MoS₂.

The authors should be more accurate when they mention "a spin transition" what do they have in mind? Is this indeed a collective order, eventually of spin glass type? It is surprising that this magnetic phase is not destroyed by thermal fluctuations.

As mentioned above the authors should have investigated the possibility of magnetic hysteresis in their data.

It is interesting to correlate the amplitude of the magnetoresistance to the magnetic

susceptibility measurements on the QDTP molecules, however the data shown in S3 on a very limited range of temperature between 375 and 400K is not convincing at all. Whereas the susceptibility only increases by 15% at most, the coefficient of the magnetoresistance changes by a factor 4...

In conclusion this work is at a too premature stage to be published. Extra checks and complementary new experiments are needed before publication.

Reviewer #2 (Remarks to the Author):

The manuscript by Datta et al reports on the observation of magneto-conductance in 2D materials (Graphene and MoS₂) induced by singlet-to-triplet transitions at high temperature in magnetic adsorbed molecules (QDTP). In particular, the MoS₂-QDTP hybrid structure is proposed as a small magnetic field detector.

I found the manuscript very interesting and of high impact for the future development of spin-based applications of novel 2D materials like MoS₂. Nevertheless, I have a comment that should be addressed to recommend its publication in Nature Comm.

The authors carry out first-principle calculations in order to show how the charge transfer between the molecule and 2D-material switch when going from Graphene to MoS₂. The calculations seem to be correct but I missed some information about the electronic structure. For instance, I would like to know if the states of the 2D material become spin-polarized due to the presence of the molecule or where the states of the molecule appear with respect to the Dirac point (magnetic proximity effects) maybe including a figure containing the PDOS. Besides, the authors only plot the charge density but do not say anything about the spin density. In my opinion, this is mandatory when the manuscript is describing physics involving spin-polarized devices.

Reviewer #3 (Remarks to the Author):

In this manuscript, authors investigated the MR of 2D materials coated with QDTP. They found that Graphene-QDTP has a large positive MR while MoS₂-QDTP demonstrates a large negative MR (~100%) above the magnetic transition of QDTP. They believe that this study suggests the possibility of room-temperature spintronics devices. Although I see some potential interest I find following problems with the main points of this study.

1. It is claimed that the positive MR observed for the Graphene/QDTP devices is due to the inhomogeneous charge impurities. The authors should provide the MR or MC curves for different gate voltages and for the field oriented in different planes. Furthermore to aid understanding it would be useful if the authors measured MR or MC plots for the pristine graphene and QDTP deposited on other materials, e.g. Cu, etc.

2. Why is the carrier mobility for the graphene/QDTP system greater than that of the pristine graphene? Since mobility is typically dependent on order within the system, and the QDTP acts a source of disorder, how do the authors explain the observed difference?

3. The authors claimed that the negative MR observed for the MoS₂/QDTP devices is due to the exchange interaction at the interface. The authors should test this supposition by inserting a non-magnetic spacer layer between the MoS₂ and QDTP to decouple the effect and also measure the MR of MoS₂/QDTP for the field oriented in different planes.

A few points relating to the text,

It would be good if the unabbreviated name of QDTP was explicitly mentioned once in the text. Alternating between the terms magnetoresistance and magnetoconductance could lead to confusion when it comes to interpreting the primary results of the paper, it might be good to standardize this throughout the text.

There are quite a number of typographical errors in the manuscript. E.g. QDTP on line 175. However most are syntactic errors, E.g. in the abstract, line 30, "Differing binding modes of QDTP causes either n or p doping of the 2D channel on MoS₂ and graphene, respectively", should probably read as: "Different binding modes of QDTP cause either n or p doping of the 2D channels of MoS₂ and graphene" The sentence from line 69 "Herein, we demonstrate a hybrid device comprising a QDTP molecular assembly which are charge coupled to 2D materials such as graphene or MoS₂." Since the manuscript explicitly deals with graphene and MoS₂ should it not read: "which are charge coupled to a 2D material, graphene or MoS₂." And the meaning of the sentence "The temperature dependence of σ_{xx} shows a similar trend as the carrier density and increasing to 8 times its RT value" from line 196 is corrupted. Particular care needs to be taken throughout the manuscript when dealing with the definite article, the indefinite article and pluralisation. Care should also be taken to avoid formatting errors such as (n_{exp} / n_{300K}) in the caption of Figure 3.

In summary, there are results here that may be of physical interest, but the presentation limits understanding-and critical analysis-of what is being measured.

Point-by-point response to **NCOMMS-16-18053A-Z**

“Magnetoresistance change in 2D materials driven by molecular spin flip” By Datta et. al.

Reviewer #1:

The authors present data on 2 different devices one with graphene and one with MoS₂ both before and after deposition of QDTP molecules very different behaviors are found for these 2 systems for the magnetoresistance after deposition of the molecules which are attributed to different charge transfer mechanisms in these 2 systems. Even if it is suggested in the text that several samples were prepared for both systems data are shown for only a single sample of graphene and MoS₂. Previous work on graphene functionalised samples with porphyrins (see Li et al. Phys. Rev. B 93, 045403 (2016)) show that charge transfer and induced magnetism strongly depend on the initial doping state of the sample.

The very broad Dirac point of the graphene sample investigated by the authors indicates a poor quality sample, with probably many dopants on the sample before deposition of the molecules.

It would be very important to present data on several better quality samples.

Response: This comment contains two parts: Firstly, the referee has concerns over the reproducibility of the charge transfer in various graphene samples. Secondly, the dependence of charge transfer and induced magnetism on the initial doping state of graphene.

To answer the first part: We have included the data (plots and statistics) and text to convince the referee about the consistency of the p-type charge transfer between QDTP and graphene (please see **modification of the draft** below). To summarize the result, we have observed hole transfer from QDTP to graphene for all graphene samples of varying mobilities, which is in accordance with our DFT calculations.

To answer the second part: We thank referee for mentioning the work of Li *et al.* (Phys. Rev. B 93, 045403 (2016)), in the mentioned work, porphyrin molecules can be both donors as well as acceptor of electrons. In both cases, after grafting, Dirac point is brought to zero gate voltage, so there is a neutralization of graphene.

In our work, **QDTP always produces hole doping**. The charge transfer does not depend on the position of the Dirac peak. Unlike the work mentioned, we choose graphene devices with Dirac peaks near to zero gate voltages, or slightly shifted to positive gate voltages (Please see the modified **supplementary information**). Please note that we have annealed our pristine graphene device at Ar-H₂ atmosphere at 200°C to remove residues and any

unintentional doping. Moreover, our work corroborates with the DFT calculations which suggests the p-type charge transfer from QDTP to graphene.

Please see data below, **we have carried out experiments** to show that the qualitative trend is similar **independent** of the original state of the graphene or MoS₂, regardless of whether high or low mobility samples were used. In the revised paper, we have substituted the old data with new device data from the high quality samples.

Response Figure 1: Transfer characteristics of 2D hybrid device. (a) Gate dependence of the two terminal conductance at room temperature for two representative graphene field-effect transistors (FET), G1 ($\mu \sim 2800 \text{ cm}^2 \text{ V}^{-1} \text{ s}^{-1}$), and G2 ($\mu \sim 23200 \text{ cm}^2 \text{ V}^{-1} \text{ s}^{-1}$); pristine graphene (black) and after deposition of QDTP molecule (red). Solid lines represents the theoretical fit of the experimental data following effective medium theory (EMT). A positive shift of charge neutrality point V_D ($\Delta V_g \sim 30V$ for G1, 17V for G2) confirms the p-type doping due to molecule. Inset: Typical Hall bar graphene - QDTP hybrid device. **(b)** Room-temperature transfer characteristic for two representative MoS₂ FETs, M1 and M2; pristine MoS₂ (black) and after deposition of QDTP molecule (red). Solid lines represents the power law fit ($\sigma \sim (V_g - V_0)^\beta$, β here is 1.29 for M1, 3.804 for M2 and V_0 is the threshold voltage) of the experimental data. Increment of σ after the deposition of QDTP molecule confirms the n-type doping of MoS₂.

According to referee's suggestion regarding statistics of measurements, the transfer curves for 5 devices have been presented in the **supplementary information (Figure S2a)** with a **table (Table T1)** containing different parameters (mobility, and doping) used to fit those curves. Texts have been added to the main draft (section: **Transfer Characteristics of hybrid device**, line 111 – 115, 127 - 133) to show the consistency of the molecule-graphene charge transfer. We have also cited the important work of **Li et al. (Phys. Rev. B 93, 045403 (2016) as [18])**

Table S1: Graphene hybrid

Sample	Mobility μ (T^{-1} or $m^2/V.s$)		Doping Δp ($10^{10} cm^{-2}$)
	Pristine	Hybrid	
Sample 1	0.28	0.68	252
Sample 2	2.32	2.51	105
Sample 3	2.91	2.88	162
Sample 4	0.42	0.44	59
Sample 5	0.58	0.71	90

The same comments are valid for the MoS2 samples.

Transfer curves of four different MoS2-QDTP devices with different mobilities have been plotted in the **supplementary Figure S2b**. We have also included texts (line 137 – 140, 143 - 146) which contain details of the samples and charge transfer in those devices.

Table S2: MoS2 hybrid

Sample	Critical exponent β	Doping Δn ($10^{10} cm^{-2}$)
Sample 1	1.29	261
Sample 2	3.80	267
Sample 3	3.10	231
Sample 4	1.03	682

Please see the **Response Figure1** below which confirms the different charge transfer for graphene-QDTP (p-type) and MoS₂-QDTP (n-type), tested for different samples. The trend is qualitatively similar over all tested samples shown (6 device data were presented below). We have also revised **Figure 2** with the better quality sample.

Data from different samples (revised Figure 2)

Response Figure 2: Room temperature Transfer curve of hybrid devices; **a** Pristine graphene and QDTP-graphene devices. A p-type charge transfer has been observed in all samples. **b** Pristine MoS₂ and QDTP-MoS₂ devices. n-type charge transfer has been observed in all samples.

Moreover, deposition of the QDTP molecules does not seem to be well controlled. Did the authors investigate different conditions of deposition of the molecules?

Response: We have used mainly two different techniques for deposition of molecules: Spin coating and drop casting. Firstly, molecules are dispersed in organic solvent, namely dichloromethane. After the characterization of the pristine device, we disperse the molecules on the graphene/MoS₂ samples in the clean room (Glove box not needed). We prefer drop casting over spin coating as later could disturb the wire bonding of the predefined pristine device. However, the observed behavior such as charge transfer, mobility enhancement are reproduced in all hybrid devices with both deposition processes, as confirmed in the room temperature probe station (where no wire bonding involved).

After mounting the sample in the cryostat, a two hour vacuum annealing at 120°C were carried out.

In order to appreciate the difference between the temperature dependence of the carrier density and magnetoresistance of coated graphene and MoS₂ samples more points in temperature are needed in Fig.3 and of course different samples.

Response: In the case of graphene-QDTP hybrid, there is no sharp jump of carrier density around the spin transition temperature, so taking more data points won't give us any new insight. We have plotted the mean magnetoconductivity and mean carrier density along with standard deviation from all measurements of graphene hybrids to show the consistency of our result.

In the case of MoS₂, we agree with the referee. Closer intervals would give us the clearer picture of the effect of the spin transition on electron transport of MoS₂. Please see the modification below.

Modification of the draft: **Data from different samples (revised Figure 3)**

Response Figure 3: Magnetoconductance (MC) and carrier density of hybrid devices; **a** Graphene-QDTP shows negative MC (black), while MoS₂-QDTP shows positive MC (red). The mean of normalized magneto conductivity with respect to zero-field value for 10 different samples from each hybrid has been plotted to show the consistency of the result. The colored region (red for MoS₂ and green for graphene) depicts the standard deviation of the data for each magnetic field. **b** Temperature dependent carrier density of the hybrid (black for graphene and green for MoS₂); MoS₂-QDTP hybrid shows a sharp jump in carrier density around spin transition temperature of QDTP (360K), while in the case of graphene-QDTP, carrier density has a monotonic decrease. Mean of normalized carrier density with respect to room temperature value has been plotted. Standard deviation for each Data point has been shown in color.

Modified Figure 3 has been added to the main draft. We have included data from 10 different samples from each hybrid (graphene-QDTP, MoS₂-QDTP) and added in the magnetoconductivity as well as in the carrier density plot. New text relating to the added data has been added in lines **152-155, 173-176**.

Magnetoconductance data is only shown for one direction of the magnetic field. It would have been useful to have also negative fields since an important question is the possible existence of magnetic hysteresis loops in this system.

Response: We thank referee for his comment. In both the hybrids, no hysteresis loop was observed (Please see modification). We must clarify that QDTP molecules are not like ferromagnetic thin film or nanoparticles with a sharp switching field of magnetization at low temperature. In the triplet state of QDTP, the spins are randomly oriented on the surface of graphene or MoS₂. The new data supports our explanation on the contrasting magnetoconductance observed in our experiment: for graphene, molecules are acting as randomly spaced inhomogeneous charge puddles, while for MoS₂, molecular spins are coupled to 2D electrons. So in both cases, there is no anisotropy involved which can produce hysteresis in the resistance.

Modification of the draft:

Hysteresis Loop (Figure S15)

Response Figure 4: Hysteresis loop as a function of field. Both hybrids (graphene-QDTP, MoS2-QDTP) shows no hysteresis in resistance as magnetic field is swept back and forth for two different temperatures, 300K (singlet state of QDTP), and 400K (triplet state of QDTP).

The authors have to be accurate in their definitions of longitudinal and Hall conductivity. They are both called ρ_{xx} (line 101).

Modification of the draft: We apologize for the typo error. We have revised this in our revised version.

Moreover the curve shown in S2 is very strange since it is very unlikely that the Hall resistance goes through a minimum at the Dirac point. It should instead change sign !. "Transverse transfer curve " in the title of the figure caption does not mean anything.

Response: We believe reviewer has overlooked the title of Hall resistance in the **supplementary Figure S4**. We have plotted modulus of Hall resistance ($|R_{xy}|$) as a function of gate voltage. So the sign change at Dirac point has been taken care of.

The authors suggest that the difference between the magnetoresistance of coated graphene and MoS₂ samples is due to larger RKKY interactions between the molecules in the triplet state on MoS₂.

The authors should be more accurate when they mention “a spin transition” what do they have in mind? Is this indeed a collective order, eventually of spin glass type? It is surprising that this magnetic phase is not destroyed by thermal fluctuations.

Response: The QDTP molecule (without any interaction with the substrate) has a spin transition at 360K from singlet to triplet state, as reported by Zeng et al. In our framework, we demonstrate a hybrid device comprising a QDTP molecular assembly which are charge coupled to 2D materials such as graphene or MoS₂. From Figure 3b ($n(T)$ vs T plot), we show that the carrier density jumps one order of magnitude in the case of MoS₂, while graphene shows monotonic decrease. So the effect of spin transition of the molecule on the 2D electron transport of the substrate is much more evident in the case of MoS₂. Here, the model we discussed is similar to diluted magnetic semiconductor like Mn-doped GaAs as reported in the reference [38], where the spins on Mn sites are coupled to conduction electrons of GaAs via exchange coupling (RKKY). In our work, the triplet spins of QDTP as non-zero spin sites are coupled to the conduction electrons of MoS₂ mediated by an exchange coupling J . Having lower carrier concentration than graphene, MoS₂ facilitates the detection of the triplet spin population in the QDTP molecule *via* spin-electron coupling.

Regarding the presence of ferromagnetic order in the QDTP molecule: The competition between an Edwards-Anderson spin glass phase and a Ferromagnetic phase has a long history in the literature. At zero temperature, the spin-glass phase always win, and at infinite temperature, thermal fluctuations would destroy any ferromagnetic order (both these scenarios were alluded to by the referee). However, looking at the classical literature [e.g. Sherrington and Southernl, J. Phy. F (1975)] we see that the Curie temperature scales as the exchange energy, which is quite large for QDTP on MoS₂. By contrast, the spin glass temperature scales as the fluctuations in the exchange energy. Since our molecules are identically produced, and our samples are high mobility with low carrier density fluctuations, we expect that our experiments are in the regime where the fluctuations in exchange energy are much smaller than their mean, and in this regime, theory predicts a stable ferromagnetic phase. Our experimental data further backs up this scenario, including the absence of hysteresis or dependence of MR with the tilting the magnetic field (as discussed below).

It is interesting to correlate the amplitude of the magnetoresistance to the magnetic susceptibility measurements on the QDTP molecules, however the data shown in S3 on a very limited range of temperature between 375 and 400K is not convincing at all. Whereas the susceptibility only increases by 15% at most,. The coefficient of the magnetoresistance changes by a factor 4...

Response: We cannot increase the temperature limit as with our Physical Property Measurement System, the highest instrument measurement temperature is measure up to 400K. Moreover, the adsorbed molecules will be evaporated at higher temperature. So we could only measure in the given window 370K – 400K. For clarity, we have recorded data in intervals of 10K in order to identify the position of the jump in carrier density, as suggested by referee.

The correlation between the magnetic susceptibility of the QDTP molecule in the spin triplet state (SQUID data) as reported by Zeng et al. and the magnetoresistance of MoS₂-QDTP hybrid from our measurement can be explained by the model we have adopted. The essence of the model is the exchange coupling between the triplet state of QDTP and the conduction electrons of MoS₂. We have simplified the equation of spin-disorder scattering resistivity to this form: $(\rho(B))/(\rho(B=0))=1-A(T)\times B^2$. Even though it qualitatively explains our results, one should note that this is not the exact equation as it does not include the exact parameters of the main equation, so direct correspondence between χT and MC would not be observed

In conclusion this work is at a too premature stage to be published. Extra checks and complementary new experiments are needed before publication.

We believe that we can convince referee #1 with the new set of experimental data included in modified draft along with explanations of the concerns raised by the referee.

Reviewer #2:

The manuscript by Datta et al reports on the observation of magneto-conductance in 2D materials (Graphene and MoS₂) induced by singlet-to-triplet transitions at high temperature in magnetic adsorbed molecules (QDTP). In particular, the MoS₂-QDTP hybrid structure is proposed as a small magnetic field detector.

I found the manuscript very interesting and of high impact for the future development of spin-based applications of novel 2D materials like MoS₂. Nevertheless, I have a comment that should be addressed to recommend its publication in Nature Comm.

Response: We thank the reviewer for reviewing our work. We are grateful to the reviewer for commenting our work as a very interesting and high-impact study for the

future development of spin-based applications of 2D materials. Below we would like to respond his/her comment in detail.

The authors carry out first-principle calculations in order to show how the charge transfer between the molecule and 2D-material switch when going from Graphene to MoS₂. The calculations seem to be correct but I missed some information about the electronic structure. For instance, I would like to know if the states of the 2D material become spin-polarized due to the presence of the molecule or where the states of the molecule appear with respect to the Dirac point (magnetic proximity effects) maybe including a figure containing the PDOS. Besides, the authors only plot the charge density but do not say anything about the spin density. In my opinion, this is mandatory when the manuscript is describing physics involving spin-polarized devices.

Response: We agree with the referee that the PDOS and spin densities are very important information for understanding spin-polarized devices. According to the reviewer's suggestions, we have performed additional calculations to analyze the partial density of states (PDOS) projected on the molecules. In addition, the spin densities of both the graphene-QDTP and MoS₂-QDTP systems are analyzed. All these information are provided in the Figure 4 in the revised manuscript and provide strong support to the model we proposed.

Modification of the draft: In the revised manuscript, Figure 4 is revised to show the LDOS and spin densities in the graphene-QDTP and MoS₂-QDTP systems. The following sentences are added in the section of "Charge transfer in hybrid devices" in discussion part (259-270).

Response Figure 5: Partial density of states (PDOS) (d) and spin density (e) for graphene–QDTP hybrid in triplet state. The dashed blue (red) lines in (d) show the position of Fermi level (Dirac point of graphene). PDOS (i) and spin density (j) for MoS₂–QDTP hybrid in triplet state. The black and red lines in (d and i) correspond to the states from the total hybrid system and the QDTP, respectively.

“The local density of states (LDOS) of the spin triplet state for both hybrid systems are shown in Fig. 4d and i, respectively. The spin-split values of the spin up and down (dn) levels of the highest occupied molecular orbital (HOMO) level of QDTP in graphene-QDTP and MoS₂-QDTP systems are 0.32 and 0.42 eV, respectively, which suggests a larger magnetic coupling in MoS₂-QDTP system, consistent with the experimental findings. A tiny amount of spin-polarization is found in the states of graphene and MoS₂ due to a weak van der Waals interaction. In Fig. 4e and j, our analysis of the spin density shows that a significant amount of the magnetic moment is localized at the dicyano groups and at the conjugated backbone while negligible spin density exists at the N-methyl group and in the host 2D materials. It is expected that intrinsic defects like vacancies and grain boundaries in graphene and MoS₂ can induce stronger interaction with the molecule, which may give rise to more efficient magnetic coupling and exchange interaction.”

Reviewer #3:

1. It is claimed that the positive MR observed for the Graphene/QDTP devices is due to the inhomogeneous charge impurities. The authors should provide the MR or MC curves for different gate voltages

Response: The referee has overlooked the main text (line 210 – 215) and **Figure S8**, where we have explicitly show MC vs T for different gate voltage regions. Maximum change of MC observed at Dirac point.

and for the field oriented in different planes.

Response: We thank referee for his/her comment. **New MC data with angle dependent magnetic field** has been added in the revised supplementary information. In fact, this study **reconfirms** our proposed model of RKKY coupling of triplet spins of QDTP and conduction electrons of MoS₂ via exchange coupling.

Graphene-QDTP devices show conventional $\cos\theta$ dependence, as also observed in graphene devices in earlier reports.

However MoS₂-QDTP hybrid device shows no field dependence while orienting in different planes. The effective Hamiltonian for the chosen model includes: (i) exchange constant J , which doesn't have any magnetic field-plane dependence, and (ii) the scalar product $S_i \cdot S_j$ which depends only on the relative orientation of the spin of QDTP (randomly spaced) and electron spin of MoS₂, not on the external magnetic field orientation. Unlike conventional ferromagnetic thin film, or magnetic nanoparticle, or for that matter molecular magnets with size or shape anisotropy, these triplet spins of QDTP are random spins on MoS₂ surface with no hard axis of magnetization. This also confirms the 'no hysteresis' result in the MC vs magnetic field loop measurement.

Modification of the draft:

As a function of the magnetic field orientation with the plane of the sample, graphene-QDTP devices show conventional $\cos\theta$ dependence similar to pristine graphene^[41] at the given temperature window (Figure S17a). However MoS₂-QDTP hybrid devices shows no dependence on the orientation of the applied magnetic field in both QDTP-singlet as well as QDTP-triplet states (Figure S17b).

New plots are added in supplementary information. Figure S17

Response Figure 6: Angle dependent magnetic field effect on conductance. Out-of-plane magnetic field is 0° , while in-plane is 90° . a graphene-QDTP hybrid has $\cos\theta$ dependence with magnetic field similar to graphene. b MoS2-QDTP has no orientation dependence with the magnetic field in different planes, for both singlet and triplet state of the molecule.

Furthermore to aid understanding it would be useful if the authors measured MR or MC plots for the pristine graphene

Response: We have mentioned that the background magneto-conductance due to presence of intrinsic defects in pristine graphene is weak ($\Delta\sigma_{xx} \sim -0.1 \mu S$) and temperature-independent in the main text (line 155-157). According to referee's comment, we have provided the data in supplementary information in the new version.

Modification of the draft:

Pristine graphene and Pristine MoS2 magnetoresistance data (Figure S18)

Response Figure 6: Magnetoresistance of pristine 2D devices used in the experiment: a No significant change is observed in the case of both devices. While graphene has weak magnetoresistance at room or higher temperature due to presence of impurities, pristine MoS₂ shows no such change with magnetic field.

QDTP deposited on other materials, e.g. Cu, etc.

Response: Using copper surface as a substrate for QDTP, as suggested by the referee, did not work as copper is a metallic surface. So we cannot see any effect of QDTP on the resistance of copper film.

2. Why is the carrier mobility for the graphene/QDTP system greater than that of the pristine graphene? Since mobility is typically dependent on order within the system, and the QDTP acts a source of disorder, how do the authors explain the observed difference?

Response: At room temperature or higher temperature, we have systematically measured the gate voltage dependence of the resistance before and after QDTP deposition (Please see supplementary table T2). In most of the graphene samples, post deposition of molecules, we observe mobility increment along with p-type doping due to charge transfer. Similar results have been reported in previous studies: (i) graphene samples functionalized with insulating nanoparticles (iron and titanium oxide, CdSe) by Deqi Wang et al., (ii) porphyrin coated graphene by Li et al. While our measurements alone cannot rule out other possibilities, this mobility increase is likely due to the compensation of charged impurities. Tallying the statistics of our graphene hybrid samples and these previously reported works we believe that similar mechanism could be also applicable. There may be possibilities that molecules ionize and neutralize charge impurities on graphene or on the silicon substrate, and therefore decrease the disorder induced scattering.

Modification of the draft: We have added one extra line (line no 130) and cited relevant papers in the revised manuscript:

“Increment of carrier mobility in graphene after the deposition of QDTP molecule can be explained by compensation of charged impurities, as also reported in the previous studies of nanoparticles coated graphene^{21,22}.”

3. The authors claimed that the negative MR observed for the MoS₂/QDTP devices is due to the exchange interaction at the interface. The authors should test this supposition by inserting a non-magnetic spacer layer between the MoS₂ and QDTP to decouple the effect and also measure the MR of MoS₂/QDTP for the field oriented in different planes.

According to the suggestion of the referee, we have fabricated new MoS₂-QDTP device with a hBN spacer layer hBN in between MoS₂ and the QDTP molecule to decouple the effect of exchange interaction. Indeed, we observe that hBN hinders the charge transfer (Figure S19) between QDTP and MoS₂. Moreover, when applying magnetic field to the MoS₂-hBN-QDTP device at 400K (triplet state of QDTP), we see no magnetoconductance. While for the same flake of QDTP/MoS₂ without hBN, we could observe 80% magnetoconductance at 400K.

This result corroborates our suggested model of exchange coupling between triplet spins of QDTP and electrons of MoS₂.

Modification of the draft: Introducing a spacer layer like hBN in between MoS₂ and the QDTP molecule decouples the magnetoconductance effect at high temperature, which explains the presence of exchange coupling in the MoS₂-QDTP device above the spin transition temperature of the molecule (Supplementary Figure S19a-b).

Response Figure 7: Decoupling exchange coupling 2D molecule hybrid. a hBN blocks the charge transfer from QDTP to MoS₂. Unlike MoS₂-QDTP, no change of transfer curve occurs after doping with QDTP in MoS₂-hBN-QDTP device. b In case of MoS₂-hBN-QDTP, no magnetoconductance is observed at 400K.

A few points relating to the text,

Response: We thank referee for highlighting the errors in text. We accept referee's concern. We have rectified and proof-read carefully.

It would good if the unabbreviated name of QDTP was explicitly mentioned once in the text.

Modification of the draft: Added on line 54: Quinoidal dithienyl perylenequinodimethane (QDTP)

Alternating between the terms magnetoresistance and magnetoconductance could lead to confusion when it comes to interpreting the primary results of the paper, it might be good to standardize this throughout the text.

Modification of the draft: We have used the term "magnetoconductance" in the discussion part in the revised manuscript.

There are quite a number of typographical errors in the manuscript. E.g. QDTP on line 175.

Modification of the draft: Changed to QDTP in the revised manuscript.

However most are syntactic errors, E.g. in the abstract, line 30, “Differing binding modes of QTDP causes either n or p doping of the 2D channel on MoS2 and graphene, respectively”, should probably read as: “Different binding modes of QTDP cause either n or p doping of the 2D channels of MoS2 and graphene”

Modification of the draft: Modified the line as “Different binding modes of QTDP cause either n or p doping of the 2D channels of MoS2 and graphene”

The sentence from line 69 “Herein, we demonstrate a hybrid device comprising a QTDP molecular assembly which are charged coupled to 2D materials such as graphene or MoS2.” Since the manuscript explicitly deals with graphene and MoS2 should it not read: “which are charge coupled to a 2D material, graphene or MoS2.”

Modification of the draft: Modified the line as “which are charge coupled to a 2D material, graphene or MoS2.”

And the meaning of the sentence “The temperature dependence of σ_{xx} shows a similar trend as the carrier density and increasing to 8 times its RT value” from line 196 is corrupted.

Modification of the draft: Modified the line as “The temperature dependence of σ_{xx} shows a similar trend as the carrier density increasing 8 times from its RT value.”

Particular care needs to be taken throughout the manuscript when dealing with the definite article, the indefinite article and pluralisation. Care should also be taken to avoid formatting errors such as (n_{exp} / n_{300K}) in the caption of Figure 3.

Modification of the draft: Modified the caption of Figure 3.

We hope we have successfully addressed all the concerns of the referees and the work can now be considered for publication in Nature Communications.

Reviewers' comments:

Reviewer #1 (Remarks to the Author):

This new version of the paper is much better than the previous one.

In particular it is now clear the effects reported have been observed in many samples with different mobilities. The experimental data showing strong modifications of the magnetoconductance with molecules coating sound now quite solid.

My only concern is related to the interpretation of the data. The authors refer to the ferromagnetic phase transition observed in GaAsMn and other magnetic semiconductors. Whereas the correlation between the increase of magnetoconductance observed magnetism in the molecules due to a thermally activated crossover between a singlet and a triplet state, I don't think that there is any sign in this paper of a spin "phase" transition as claimed by the authors. The temperature transition should be higher than 400K which is a very strong unproven statement.

It is likely that the exchange interaction between conduction electrons and localized triplet spins on the molecules play an important role in the physical phenomena described here. However there is no quantitative estimation of the amplitude of this exchange coupling neither on the amplitude of the RKKY spin-spin interactions between neighboring molecules in this system. I do not see any signature of long range order indicating a ferromagnetic phase transition for MoS₂ coated samples as claimed by the authors in their response letter.

If we assume, as seems more reasonable, that the molecular spins are not strongly coupled within one another the spin polarization induced by a 50mT magnetic field is on the order of 1% or less and cannot explain alone the large positive magneto-conductance observed in coated MoS₂.

On the other hand the literature on organic conductors is full of data showing large positive or negative low field magnetoconductance at room temperature which is very similar to what is observed here, see for example <https://arxiv.org/pdf/cond-mat/0409753> and Bobbert et al. PRL 99, 216801 (2007).

Without more quantitative information on their system, the authors should avoid to us the terminology spin"phase" transition, "spin transition" only is acceptable.

With these modifications I consider that now the paper can be published in Nature Communications.

Reviewer #2 (Remarks to the Author):

The authors have made a great effort improving the manuscript and most of the questions raised by the referees have been properly addressed in the last version. However, in my opinion, a piece of physics is still missing when describing the magneto-conductance (MC) in the MoS₂-QDTP hybrid system.

The MC in the Graphene-QDTP heterostructure is supported by the fact that there is a charge transfer from Graphene to QDTP leading to electron-hole charge puddles in the Graphene layer (disorder). Theoretical calculations show that the charge transfer increases at high temperature due to the singlet-to-triplet spin transition in the QDTP affecting the MC. However, first-principles calculations on MoS₂-QDTP do not show an appreciable change in charge transfer when going from singlet to triplet state (page 11 of the manuscript). This result contrasts with the huge change in carrier concentration reported in Fig. 3(b) at the transition temperature. The authors argue in pages 12 and 13 that the mechanism at play is based on magnetic scattering and they make reference to the Wigner-Seitz radius and RKKY interaction between spin state of the QDTP and electrons spin in MoS₂. This puzzle me because the spin density plotted in Fig. 4(j) do not show any appreciable weight in the MoS₂ atomic structure. This means that the injected current may not be strongly affected by the spin state of the molecules. At least, to show such a huge difference in the MC.

I think addressing this question is key to properly describe the magneto-transport properties of MoS₂-QDTP heterostructures.

In summary, I think that the manuscript has been nicely improved but there is still an important question to be addressed in order to deserve publication in Nature Comm.

Reviewer #3 (Remarks to the Author):

After reviewing both the rebuttal and the latest version of the manuscript I would like to thank the authors for their thoughtful responses. However, I am still not convinced of the explanation about the transport measurement, especially for MoS₂/QTDP hybrid system.

The author observed a large negative MR in MoS₂/QTDP hybrid system at a temperature above 370K (Figure S7) and explained their results based on the large exchange coupling between carriers in MoS₂ and the spin-triplet state of QTDP. The MR curves are so different above and below 370 K. So, they cited Ref 9 to support their explanation that QTDP has a spin transition from the singlet to the triplet state at 360K. However, I cannot find this claim in Ref. 9. If authors really want to claim that, they should measure the M-H loops at different temperatures or M-T curve for QTDP.

Secondly, authors try to explain the observed negative MR based on spin-disorder scattering and try to fit it with $1-A(T) \times B^2$ only in very small field region. However, MR only changes slightly above 200 mT.

Thirdly, authors performed dft calculation to support their explanation. They should at least try to calculate J and compare it with experimental data.

At this moment, I cannot recommend the manuscript for publication in Nature Communications until these concerns are adequately resolved.

Referee #1:

1. Whereas the correlation between the increase of magnetoconductance observed magnetism in the molecules due to a thermally activated crossover between a singlet and a triplet state, I don't think that there is any sign in this paper of a spin "phase" transition as claimed by the authors. The temperature transition should be higher than 400K which is a very strong unproven statement.

We thank the referee for pointing this out. We have to clarify that the exchange couplings is between the spins in the molecules and the electrons in MoS₂. Not between the molecules themselves. The only place where we have used "spin phase transition" is on page 13, line 299. To avoid misunderstanding, we have removed the phrase in the revised manuscript. To elaborate, we modelled the addition of QDTP to MoS₂ similar to dilute magnetic semiconductors with the addition of Mn atoms to GaAs. Following PRB 57 R2037 (1998) we modelled the changes in magnetoresistance as the reduction in scattering when the spins in the magnetic molecules are aligned to the external magnetic field, and the quantitative measure of the MR gives us an estimate of the exchange coupling J between the QDTP spins and conduction electrons in MoS₂. Our measured value of $J = 200 \text{ eV \AA}^3$ (see revised text) is slightly larger, but comparable to the values seen in dilute magnetic semiconductors (the above paper finds 130 eV \AA^3). But as pointed out by the referee, this analysis applies to the high-temperature phase $T > T_c$. The referee is right that we see no experimental evidence of a spin phase transition, but we can use our experimentally determined value of J to speculate that if there was no magnetic transition in the QDTP and it remained in a triplet state to lower temperatures, then we would expect a spin phase transition at 170 K. But we do not see this transition because the molecule undergoes the singlet-triplet transition at 360 K. In this revised version we have been careful to **avoid claiming to have observed a spin phase transition.**

2. On the other hand the literature on organic conductors is full of data showing large positive or negative low field magnetoconductance at room temperature which is very similar to what is observed here, see for example <https://arxiv.org/pdf/cond-mat/0409753> and Bobbert et al. PRL 99, 216801 (2007). Without more quantitative information on their system, the authors should avoid to use the terminology spin “phase” transition, “spin transition” only is acceptable.

First, as discussed above, in the revised version **we avoid the terminology “spin phase transition”**. Second, we thank the referee for pointing out the two references on organic conductors (we have added these as references in our introduction), although we do not think the physics there is directly relevant to our case. In the first paper, T.L. Francis et al. New Journal of Physics 6 (2004) 185 (and arXiv:0409753) conclude: “To the best of our knowledge, the discovered effect is not adequately described by any of the MR mechanisms known to date”. They seem to think that weak localization might be at play, but since we know that the electron phase coherence in MoS₂ is destroyed at these temperatures [see: *Phys. Rev. Lett.* **116**, 046803 (2016)], we can effectively rule this out as well. We agree with those authors that none of the explanations they considered matches either their experiment or ours. As for Bobbert et al. PRL 99, 216801 (2007) from the same group, they propose a bipolaron mechanism for organic magnetoresistance where they have no net spin polarization, but correlations between out-of-equilibrium and equilibrium spins exposed to a classical randomly fluctuating local hyperfine field. If our mechanism had polaronic origin, it would occur without the addition of the magnetic molecules (it doesn’t), and have a strong temperature dependence (it doesn’t). There is also more than an order of magnitude difference in the magnetic field scale (5 mT in OMAR and 100 mT in our experiment), and a factor of 3 difference in temperature scale. It seems clear that in our case, the magnetic molecules create magnetic sites, and squid measurements show polarizability, providing a very different mechanism than the one at play in organic conductors.

3. There is no quantitative estimation of the amplitude of this exchange coupling neither on the amplitude of the RKKY spin-spin interactions between neighboring molecules in this system. I do not see any signature of long range order indicating a ferromagnetic phase transition for MOS₂ coated samples as claimed by the authors in their response letter.

Both Referee #1 and Referee #3 asked for a quantitative estimate for the amplitude of the exchange coupling. In the first dilute magnetic semiconductors, PRB 57 R2037 (1998), they found exchange couplings $J = (150 \pm 40) \text{ eV \AA}^3$, although theoretical proposals in Science 287 1019 (2000) suggest how to obtain much larger couplings. We can determine these values from DFT calculations. The energy difference between singlet QDTP/MoS₂ and triplet QDTP/MoS₂ is 0.36 eV over a length scale of approximately 2 nm (in the DFT calculation). This gives a value of J of approximately 140 eV \AA^3 , which is comparable to the first DMS studies (and within 40 percent from the experimental value determined above). We note that calculations of J for just the QDTP without the MoS₂ conduction electrons is a factor of 1000 smaller, which is further evidence for the model presented here.

4. If we assume, as seems more reasonable, that the molecular spins are not strongly coupled within one another the spin polarization induced by a 50mT magnetic field is on the order of 1% or less and cannot explain alone the large positive magneto-conductance observed in coated MOS₂.

We agree with the referee that the molecular spins are not strongly correlated. But there are two factors. First, these molecules have very tiny coercive field as these are not like d-electron systems. So the spin state of molecule can be aligned by very small magnetic field. Secondly, when we discuss about magnetoconductance, the exchange coupling (140eV) between molecular spin of QDTP and electron spin of MoS2 comes into play. This value is three orders of magnitude larger than the spin-spin coupling between two molecules (100meV). We believe the observed large magnetoconductance at small field (50mT) is due to these two key factors.

Reviewer #2:

The authors have made a great effort improving the manuscript and most of the questions raised by the referees have been properly addressed in the last version. However, in my opinion, a piece of physics is still missing when describing the magneto-conductance (MC) in the MoS2-QDTP hybrid system.

The MC in the Graphene-QDTP heterostructure is supported by the fact that there is a charge transfer from Graphene to QDTP leading to electron-hole charge puddles in the Graphene layer (disorder). Theoretical calculations show that the charge transfer increases at high temperature due to the singlet-to-triplet spin transition in the QDTP affecting the MC. However, first-principles calculations on MoS2-QDTP do not show an appreciable change in charge transfer when going from singlet to triplet state (page 11 of the manuscript). This result contrasts with the huge change in carrier concentration reported in Fig. 3(b) at the transition temperature. The authors argue in pages 12 and 13 that the mechanism at play is based on magnetic scattering and they make reference to the Wigner-Seitz radius and RKKY interaction between spin state of the QDTP and electrons spin in MoS2. This puzzle me because the spin density plotted in Fig. 4(j) do not show any appreciable weight in the MoS2 atomic structure. This means that the injected current may not be strongly affected by the spin state of the molecules. At least, to show such a huge difference in the MC.

I think addressing this question is key to properly describe the magneto-transport properties of MoS2-QDTP heterostructures.

In summary, I think that the manuscript has been nicely improved but there is still an important question to be addressed in order to deserve publication in Nature Comm.

Reply: We thank the referee for this insightful comment. As inspired by the referee's suggestions, we reexamined our model to understand the underlying cause of the enhanced carrier concentration in QDTP decorated MoS2 case. We realize we have neglected to take into account the presence of sulfur vacancies on MoS2 previously and this has profound consequence on the charge transfer and spin interactions. In the revised manuscript, we have to take into consideration the presence of sulfur vacancies (V_s) which are well known as electron trap sites.

Figure P1. PL spectrum for pristine MoS₂ and hybrid QTDP-MoS₂. The PL was taken at 300 K air with a constant laser excitation power. Inset: Raman spectrum of the same flake. The PL intensity was higher for QTDP-MoS₂ even though QTDP molecule does not show PL in the measured range, the enhanced PL was due to the passivation of electron trap sites on MoS₂ by QTDP [see reference 32].

Passivation of these sulfur vacancies by molecular adsorbates has been found to increase carrier densities and mobilities. [Please see reference 32 Newaz, A. K. M. *et al. Solid State Commun.* **155**, 49–52 (2013), Mak, K. F. *et al. Nature Mater.* **12**, 207–211 (2012), Tongay, S. *et al. Sci. Rep.* **3**, 2657 (2013)].

This is indeed reflected in our case when we observe enhanced photoluminescence (Figure P1) from MoS₂ following the adsorption of these molecules (control studies show that these do not show PL). The concentration of these sulfur vacancies (V_s), which are ubiquitous in MoS₂, has been estimated to be 10^{10} cm^{-2} by previous authors [Tongay, S. *et al. Sci. Rep.* **3**, 2657 (2013)]. In other words, this is equivalent to one defect per 10^5 Mo atom (Mo atom density $\sim 10^{15} \text{ cm}^{-2}$). That is why we can observe a large change in carrier density (10 times) when the molecule interact with these defect sites of MoS₂ by transferring electrons to it.

Taking into account the presence of sulfur vacancies, we have performed DFT calculations to examine the spin density of the hybrid system with QTDP adsorbed above the V_s on MoS₂ sheet. Charge transfer between the QTDP and MoS₂, as plotted in Fig. P2, shows that the triplet-state adsorption has a much larger charge transfer (0.204 electrons) than that of the singlet-state adsorption (0.166 electrons). This can be appreciated by the fact that unpaired electrons on the triplet and on the S vacancies interact strongly. As shown by the plot of spin density in Fig. P3, there is indeed appreciable weight of spin density in the MoS₂ sheet. This spin density in the MoS₂ can be due to the asymmetric coupling of the V_s related localized defective states with that of the spin up and down components of the triplet state molecule (Fig. P4), while such spin density and asymmetric coupling are absent in the singlet state molecule.

Figure P2. The amount of transferred charge up to z point $\Delta Q(z)$ for QDTP adsorbed above the V_S contained MoS_2 . The horizontal dashed line defines the maximum amount of transferred charge from the QDTP to MoS_2 . The amount of transferred charge from QDTP in singlet and triplet state to MoS_2 are 0.166 and 0.204 e, respectively.

Figure P3. Spin density of CF_3 QDTP in the triplet state adsorbed above the V_S contained MoS_2 .

Figure P4. LDOS for the QDTP in singlet and triplet states adsorbed above the V_5 contained MoS_2 . The black and red lines correspond to the total and local density of states (DOS) of the QDTP adsorbed MoS_2 and QDTP, respectively. The a_1 is the V_5 related defective levels in the band gap of MoS_2 .

Action in the revised paper:

In the revised manuscript, the Fig. 4 is replotted and in the end of “Charge transfer in hybrid devices” section in the discussion part is rewritten as follows:

“ In the case of the hybrid QDTP- MoS_2 system, we attribute the enhancement of the carrier density to a synergistic effect of the QDTP molecule and the sulfur vacancy (V_5) of the MoS_2 . It is well known that MoS_2 can readily form intrinsic defects like the V_5 and grain boundaries unintentionally during growth and device fabrication. In principle these defects can have unpaired spins and thus have a stronger interaction with spin molecules. . Charge transfer between the QDTP and MoS_2 plotted in Fig. 4h indeed shows that the triplet-state (0.21 e/molecule) adsorption has a much larger charge transfer than that of the singlet-state (0.16 e/molecule) adsorption.

Our theoretical calculations show that the different charge transfer behaviour observed between graphene and MoS_2 has its origin in the different interactions of these 2D layers with the functional groups in the QDTP. The isosurface plot of the differential charge density reveals that the dicyano groups in QDTP act as electron acceptors with respect to graphene (Fig. 4a,b), whereas, the N-methyl (NCH_3) group located in the in the mid-section of the molecule acts as an electron donor when present on the surface of MoS_2 (Fig. 4f,g).

The local density of states (LDOS) of the spin triplet state for both hybrid systems are shown in Figure 4d and i, respectively. A tiny amount of spin-polarization is found in the states of graphene and MoS_2 due to a weak van der Waals interaction with QDTP. In Fig. 4e and j, our analysis of the spin density shows that significant amount of the magnetic moment is concentrated at the dicyano groups and at the conjugated backbone. As shown by the plot of spin density in Fig. 4j, there is indeed appreciable weight of spin density in the MoS_2 sheet. The enhanced spin density in MoS_2 is due to the asymmetric coupling of the V_5 -related localized defective states with that of the unpaired spins of the triplet state molecule, while such spin density and asymmetric coupling are absent in the singlet state molecule (Fig. 4i). ”

Reviewer #3:

After reviewing both the rebuttal and the latest version of the manuscript I would like to thank the authors for their thoughtful responses. However, I am still not convinced of the explanation about the transport measurement, especially for MoS₂/QTDP hybrid system.

The author observed a large negative MR in MoS₂/QTDP hybrid system at a temperature above 370K (Figure S7) and explained their results based on the large exchange coupling between carriers in MoS₂ and the spin-triplet state of QTDP. The MR curves are so different above and below 370 K. So, they cited Ref 9 to support their explanation that QTDP has a spin transition from the singlet to the triplet state at 360K. However, I cannot find this claim in Ref. 9. If authors really want to claim that, they should measure the M-H loops at different temperatures or M-T curve for QTDP.

Figure P5. Left figure is taken from Ref. 9 which is the first claim of this transition in the literature. Right-hand figure is our data for the samples presented in the manuscript.

We cited Reference 9 for this data because they were the first to make the claim of a singlet-triplet transition at around 370 Kelvin. This can be found in Figure 3 (a) (accessible through DOI 10.1039/C4SC00659C). However, we agree with the Referee that the data in that paper is not so clear. Therefore, we have presented our own measurements on the samples (right panel Figure P5) and the data shows a clear spin transition (increase in spin signal $\chi_{mol} T$) around 360K, as indicated by an arrow.

Action: The new data is now included in Figure 1b.

Secondly, authors try to explain the observed negative MR based on spin-disorder scattering and try to fit it with $1-A(T)xB^2$ only in very small field region. However, MR only changes slightly above 200 mT.

Allow us to make this clearer, first, as can be seen in the upper data of Figure 3 (a), for MoS₂, the change in magnetoconductance occurs in the range between 0T and 100 mT (we show the data out to 200 mT). For fields larger than 200 mT there is no appreciable change in magnetoconductance. One can expect saturation occurs at 200 mT because at this field all the spins on the QDTP molecules are aligned with the external magnetic field (and at even larger fields, the spins are frozen out, and thus there would be no effect on the magnetoconductance). The interesting regime is at the low-field magnetic regime

where spin on the molecules can interact with MoS₂ electrons. According to the remarks, the Referee is referring to supplementary figure S13, where we show the low field data (which is quadratic in field) between zero and 50 mT. This is about half the total range where we see any variation, and the range in which the variation is quadratic (expected in the low field limit). Our approximation for determining of A(T) should be considered valid only in the limit of B → 0.

Thirdly, authors performed dft calculation to support their explanation. They should at least try to calculate J and compare it with experimental data.

Reply: As suggested by the referee, we have performed additional calculations to evaluate the coupling constant J with using the energetic difference (0.36 eV) between the spin triplet and singlet states. The theoretical values of J based on the current structural model is estimated to be around 131 eV A^{o2}. It is expected that this value is sensitive to the packing configurations of the adsorbing QDTP molecules above the 2D sheets. We have also estimated J to be 200 eV A³ from the experiment, following equation (3) of PRB 57 R2037 (1998):

$$J = \left[\frac{1}{2\pi^2} \rho \frac{n e^2 h^3}{k_F m^2 n_s} \frac{1}{[S(S+1) - \langle S \rangle^2]} \right]^{1/2}$$

In our case, we have used S = 1, n is the electron density in MoS₂ which is estimated to be 0.6x10²¹ e/cm², n_s is the doping concentration of the molecule which is estimated to be around 0.27x10¹⁸ m⁻², m is the effective mass ~ 0.4m_e, and <S> calculated from the SQUID measurement. It is noted that the n_s can vary up to orders depending on the doping concentration and the packing density of the molecules. Thermal average of <S> can be found from the SQUID measurement.

We have also estimated J (considering intermolecular spin interaction) based on experimental measurement of the susceptibility of the molecules, and it turns out to be 100meV, much smaller than spin-electron exchange coupling (140 eV).

Action: These additional calculations are included in the discussion section, and supplementary section.

Reviewers' comments:

Reviewer #1 (Remarks to the Author):

In this new version of the manuscript authors acknowledge the absence of long range magnetic order between the QDTP molecules. In this condition I do not see how a field of the order of 0.1 T can align the magnetic moments of the molecules at the temperature of the experiments which is more than 350K! Therefore I do not understand the statement in their answer: "these molecules have a very tiny coercive field... spin state of molecule can be aligned by a very small magnetic field ". To my knowledge the relevant energy scale is the Zeeman energy which is much below 1K for individual molecular spins. I do not understand what the authors have in mind when they say in their answer to referee 3 that the spins of the QDTP molecules are aligned with the external field! I only understand that the large exchange field between conduction electrons and QDTP spins is very large which certainly suppresses spin-flip scattering for the conduction electrons even at zero magnetic field. The field dependence of the resistance still needs to be explained. In this context I cannot accept this new version of the paper.

Reviewer #2 (Remarks to the Author):

In my opinion, the questions raised by the referees have been properly addressed by the authors, and the manuscript has been improved accordingly. Therefore, I think the last version of the manuscript is ready to be published in Nat. Comm.

Reviewer #3 (Remarks to the Author):

The authors have addressed my concerns and suggestions. I have no further major objections.

Detailed Reply to Referee #1

We thank Referee #1 for reading our manuscript three times, and in the previous round recommended publication in Nature Communications. To summarize the Referee's concern, he makes the following argument: (i) A magnetic field of 0.1 Tesla has a Zeeman energy of less than 1 Kelvin. (ii) The experimental temperature is more than 350 K. (iii) Therefore, there can be no alignment of magnetic moments with the magnetic field, or zero spin-imbalance, (iv) Therefore, there can be no role of magnetic field on the resistance, and (v) Therefore, the field dependence of the resistance remains unexplained.

First, we state categorically, and for the record, that the non-zero spin imbalance at temperatures greater than 360 K is **an empirical fact**.

We know this because we directly measure the magnetic moments using SQUID magnetometry. These superconducting quantum interference device measurements were done with the powder sample between 2K and 380 K. (The XT vs T measurement done at $B = 0.5T$, is shown Figure 1 of the main text). In addition, we note that the existence of a significant amount of thermally populated triplet biradical species (which have slightly higher energy than the singlet diradical) is also observed in ESR spectra, as was previously reported in Ref. [10], where the open-shell diradical nature of the molecule was confirmed from the electron spin resonance signal. At temperatures above the triplet-singlet transition, we find experimentally that the magnetic susceptibility increased. At high temperature (370K), we observe a magnetic moment of 0.02 emu/mol. From the SQUID data, the fraction of spins, which is 2% at 370K, can be calculated. These experimental data show that there exists a non-zero spin imbalance at high temperature in the molecule, however small it might be. These are just statements of **what we observe experimentally**, and they coincide with similar SQUID measurements done by Ref. 10 available at DOI: 10.1039/c4sc00659c. So claim (iii) by the referee is incompatible with our observations.

We can calculate the thermal average of spin $\langle S \rangle$ within a Boltzmann picture and show that is consistent with the SQUID measurements. Please see the attached figure below. Since it is correct that temperature is much larger than Zeeman, this value is small (e.g. $\sim 10^{-3}$ in the black curve at 360K), but non-zero. For the $B=0.5$ T, used in the SQUID we would estimate the fraction of aligned spins to be $\sim 0.5\%$, which matches with our experimental estimate of fraction of spins being 2%. So, the Boltzmann estimate is reasonable.

[Details about calculation:

$$\langle S \rangle = \frac{\sum m_s e^{x m_s}}{\sum e^{x m_s}}$$

, where $m_s = -1, 0, 1$, and $x = g \mu_B B / (k_B T)$, with $g = 2$, and the μ_B here is the usual μ_B^* (m_e/m^*). After simplification, it is just $\langle S \rangle = 2 \sinh(x) / (1 + 2 \cosh(x))$]

To calculate the effect on resistivity, we need to multiply the difference $[S(S+1) - \langle S \rangle^2]$ by J^2 . As the referee acknowledges, since J is large (more than 1000K, J^2 is a very large number i.e. $> 10^6$). Even if the difference is small, the product of $J^2 * [S(S+1) - \langle S \rangle^2]$, is not small, and is what gives rise to our magnetoresistance. Moreover, 10^{-4} change in the J^2 term gives a one order change to resistance, as observed in experiment **[Supplementary Figure S20]**. We note that the large value of J is also consistent with what was observed in Ref. 10 available at DOI: 10.1039/c4sc00659c.

So, to conclude, while claims (i) and (ii) are correct, claim (iii) should be that the "alignment is small" (but it is something we measure directly in our experiment). Claim (iv) is incorrect since our exchange energy is so large, and regarding (v) the fact is that we can apply the theory of spin-dependent scattering used in dilute magnetic semiconductors to agree with our observations by using a single adjustable parameter that we determine experimentally by mapping the quadratic co-efficient of resistivity to our observed spin imbalance in the SQUID measurement, which suggests that our understanding of the system is rather robust. Therefore, we believe the error in the referee's analysis was that he was taking 10^{-4} to be equal to zero, which contradicts what we see in our experiments.

Please see below the plot we have generated:

Reply Figure 1: Plot of thermal average of spin ($\langle S \rangle$), and $J^2 [S(S+1) - \langle S \rangle^2]$ as a function of temperature (T), as calculated considering Boltzmann statistics. Qualitatively, a smaller fraction of $\langle S \rangle$ could contribute a large MR, if the exchange interaction (J) is large, as happened in our hybrid MoS_2 device.

With this, we hope we have fully answered the referee and that our work can now be published in Nature Communications.